# Learning Linear Causal Representations from Interventions under General Nonlinear Mixing

**Simon Buchholz**[1*]  **Goutham Rajendran**[2*]  **Elan Rosenfeld**[2]

**Bryon Aragam**[3]  **Bernhard Schölkopf**[1]  **Pradeep Ravikumar**[2]

[1]Max Planck Institute for Intelligent Systems, Tübingen, Germany
[2]Carnegie Mellon University, Pittsburgh, USA
[3]University of Chicago, Chicago, USA

## Abstract

We study the problem of learning causal representations from unknown, latent interventions in a general setting, where the latent distribution is Gaussian but the mixing function is completely general. We prove strong identifiability results given unknown single-node interventions, i.e., without having access to the intervention targets. This generalizes prior works which have focused on weaker classes, such as linear maps or paired counterfactual data. This is also the first instance of identifiability from non-paired interventions for deep neural network embeddings and general causal structures. Our proof relies on carefully uncovering the high-dimensional geometric structure present in the data distribution after a non-linear density transformation, which we capture by analyzing quadratic forms of precision matrices of the latent distributions. Finally, we propose a contrastive algorithm to identify the latent variables in practice and evaluate its performance on various tasks.

## 1 Introduction

Modern generative models such as GPT-4 [59] or DDPMs [26] achieve tremendous performance for a wide variety of tasks [7]. They do this by effectively learning high-level representations which map to raw data through complicated *non-linear maps*, such as transformers [89] or diffusion processes [79]. However, we are unable to reason about the specific representations they learn, in particular they are not necessarily related to the true underlying data generating process. Besides their susceptibility to bias [20], they often fail to generalize to out-of-distribution settings [5], rendering them problematic in safety-critical domains. In order to gain a deeper understanding of what representations deep generative models learn, one line of work has pursued the goal of causal representation learning (CRL) [75]. CRL aims to learn high-level representations of data while simultaneously recovering the rich causal structure embedded in them, allowing us to reason about them and use them for higher-level cognitive tasks such as systematic generalization and planning. This has been used effectively in many application domains including genomics [52] and robotics [43, 94].

A crucial primitive in representation learning is the fundamental notion of identifiability [37, 70], i.e., the question whether a unique (up to tolerable ambiguities) model can explain the observed data. It is well known that because of non-identifiability, CRL is impossible in general settings in the absence of inductive biases or additional information [29, 50]. In this work, we consider additional information

---

*Equal Contribution

37th Conference on Neural Information Processing Systems (NeurIPS 2023).

in the form of interventional data [75]. It is common to have access to such data in many application domains such as genomics and robotics stated above (e.g. [17, 58, 57]). Moreover, there is a pressing need in such safety-critical domains to build reliable and trustworthy systems, making identifiable CRL particularly important. Therefore, it's important to study whether we can and also how to perform identifiable CRL from raw observational and interventional data. Here, identifiability opens the possibility to provably recover the true representations with formal guarantees. Meaningfully learning such representations with causal structure enables better interpretability, allows us to reason about fairness, and helps with performing high-level tasks such as planning and reasoning.

In this work, we study precisely this problem of causal representation learning in the presence of interventions. While prior work has studied simpler settings of linear or polynomial mixing [84, 88, 2, 71, 11], we allow for general non-linear mixing, which means our identifiability results apply to complex real-world systems and datasets which are used in practice. With our results, we make progress on fundamental questions on interventional learning raised by [75].

Concretely, we consider a general model with latent variables $Z$ and observed data $X$ generated as $X = f(Z)$ where $f : \mathbb{R}^d \to \mathbb{R}^{d'}$ is an arbitrary non-linear mixing function. We assume $Z$ satisfies a Gaussian structural equation model (SEM) which is unknown and unobserved. A Gaussian prior is commonly used in practice (which implies a linear SEM over $Z$) and further, having a simple model for $Z$ allows us to learn useful representations, generate data efficiently, and explore the latent causal relationships, while the non-linearity of $f$ ensures universal approximability of the model. We additionally assume access to interventional data $X^{(i)} = f(Z^{(i)})$ for $i \in I$ where $Z^{(i)}$ is the latent under an intervention on a single node $Z_{t_i}$. Notably, we allow for various kinds of interventions, we don't require knowledge of the targets $t_i$, and we don't require paired data, (i.e., we don't need counterfactual samples from the joint distribution $(X, X^{(i)})$, but only their marginals). Having targets or counterfactual data is unrealistic in many practical settings but many prior identifiability results require them, so eliminating this dependence is a crucial step towards CRL in the real world.

**Contributions**   Our main contribution is a general solution to the identifiability problem that captures practical settings with non-linear mixing functions (e.g. deep neural networks) and unknown interventions (since the targets are latent). We study both perfect and imperfect interventions and additionally allow for shift interventions, where perfect interventions remove the dependence of the target variable from its parents, while imperfect interventions (also known as soft interventions) modify the dependencies. Below, we summarize our main contributions:

1. We show identifiability of the full latent variable model, given non-paired interventional data with unknown intervention targets. In particular, we learn the mixing, the targets and the latent causal structure.

2. Compared to prior works that have focused on linear/polynomial $f$, we allow non-linear $f$ which encompasses representations learned by e.g. deep neural networks and captures complex real-world datasets. Moreover, we study both imperfect (also called soft) and perfect interventions, and always allow shift interventions.

3. We construct illustrative counterexamples to probe tightness of our assumptions which suggest directions for future work.

4. We propose a novel algorithm based on contrastive learning to learn causal representations from interventional data, and we run experiments to validate our theory. Our experiments suggest that a contrastive approach, which so far has been unexplored in interventional CRL, is a promising technique going forward.

## 2   Related work

Causal representation learning [75, 74] has seen much recent progress and applications since it generalizes and connects the fields of Independent Component Analysis [13, 28, 30], causal inference [81, 61, 63, 64, 82] and latent variable modeling [40, 3, 78, 96, 32]. Fundamental to this approach is the notion of identifiability [36, 15, 93]. Due to non-identifiability in general settings without inductive biases [29, 50], prior works have approached this problem from various angles — using additional auxiliary labels [36, 27, 6, 76]; by imposing sparsity [75, 42, 54, 103]; or by restricting the function classes [41, 9, 104, 22]. See also the survey [32]. Moreover, a long line of works have proposed practical methods for CRL (which includes causal disentanglement as a special case),

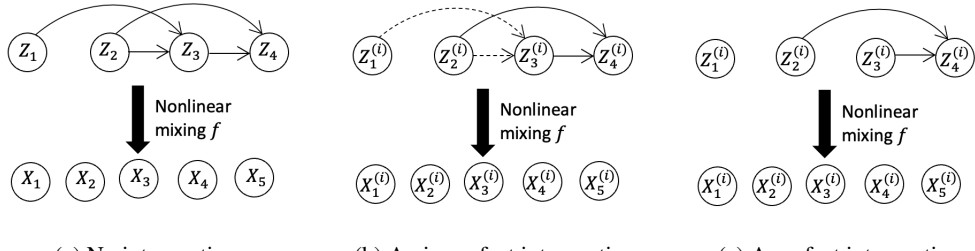

(a) No interventions      (b) An imperfect intervention      (c) A perfect intervention

Figure 1: Illustration of an example latent variable model under interventions. $(a)$ No interventions. $(b)$ An imperfect intervention on node $t_i = 3$. Dashed edges indicate the weights could have potentially been modified. $(c)$ A perfect intervention on node $t_i = 3$.

[19, 95, 16, 97, 44, 14] to name a few. It's worth noting that most of these approaches are essentially variants of the Variational Autoencoder framework [39, 68].

Of particular relevance to this work is the setting when interventional data is available. We first remark that the much simpler case of fully observed variables (i.e., no latent variables), has been studied in e.g. [23, 62, 83, 34, 18] (see also the survey [82]). In this work, we consider the more difficult setting of structure learning over latent variables, which have been explored in [46, 42, 6, 104, 1, 84, 88, 2, 71, 11, 72]. Among these, [46] assumes that the intervention targets are known, [42] specifically consider instance-level pre- and post- interventions for time-series data and [6, 104, 1] assumes access to paired counterfactual data. In contrast, we assume unknown targets and work in general settings with non-paired interventional data, which is important in various real-world applications [85, 99]. The work [48] require several graphical restrictions on the causal graph and also require $2d$ interventions, while we make no graphical restrictions and only require $d$ interventions (which we also show cannot be improved under our assumptions). [71, 11, 84, 88] consider linear mixing functions $f$ whereas we study general non-linear $f$. Finally, [2] consider polynomial mixing in the more restricted class of deterministic do-interventions. Several concurrent works study related settings [100, 35, 65, 45, 92]. For a more detailed comparison to the related works, we refer to Appendix C.

Our proposed algorithm is based on contrastive learning. Contrastive learning has been used in other contexts in this domain [31, 27, 55, 104, 91] (either in the setting of time-series data or paired counterfactual data), however the application to non-paired interventional settings is new to the best of our knowledge.

**Notation** In this work, we will almost always work with vectors and matrices in $d$ dimensions where $d$ is the latent dimension, and we will disambiguate when necessary. For a vector $v$, we denote by $v_i$ its $i$-th entry. Let Id denote the $d \times d$ identity matrix with columns as the standard basis vectors $e_1, \ldots, e_d$. We denote by $N(\mu, \Sigma)$ the multivariate normal distribution with mean $\mu$ and covariance $\Sigma$. For a set $C$, let $\mathcal{U}(C)$ denote the uniform distribution on $C$. For two random variables $X, X'$, we write $X \overset{\mathcal{D}}{=} X'$ if their distributions are the same. We denote the set $\{1, \ldots, d\}$ by $[d]$. For permutation matrices we use the convention that $(P_\omega)_{ij} = \mathbf{1}_{j=\omega(i)}$ for $\omega \in S_d$. We use standard directed graph terminology, e.g. edges, parents. When we use the term "non-linear mixing" in this work, we also include linear mixing as a special case.

## 3 Setting: Interventional Causal Representation Learning

We now formally introduce our main settings of interest and the required assumptions. We assume that there is a latent variable distribution $Z$ on $\mathbb{R}^d$ and an observational distribution $X$ on $\mathbb{R}^{d'}$ given by $X = f(Z)$ where $d \leq d'$ and $f$ is a non-linear mixing. This encapsulates the most general definition of a latent variable model. As the terminology suggests, we observe $X$ in real-life, e.g. images of cats and $Z$ encodes high-level latent information we wish to learn and model, e.g. $Z$ could indicate orientation and size.

**Assumption 1** (Nonlinear $f$). *The non-linear mixing $f$ is injective, differentiable and embeds into $\mathbb{R}^{d'}$.*

Such an assumption is standard in the literature. Injectivity is needed in order to identify $Z$ from $X$ because otherwise we may have learning ambiguity if multiple $Z$s map to the same $X$. Differentiability is needed to transfer densities. Note that Assumption 1 allows for a large class of complicated nonlinearities and we discuss in Appendix E why proving results for such a large class of mixing functions is important.

Next, we assume that the latent variables $Z$ encode causal structure which can be expressed through a Structural Causal Model (SCM) on a Directed Acyclic Graph (DAG). We focus on linear SCMs with an underlying causal graph $G$ on vertex set $[d]$, encoded by a matrix $A$ through its non-zero entries, i.e., there is an edge $i \rightarrow j$ in $G$ iff $A_{ji} \neq 0$.

**Assumption 2** (Linear Gaussian SCM on Latent Variables). *The latent variables $Z$ follow a linear SCM with Gaussian noise, so*

$$Z = AZ + D^{1/2}\epsilon \tag{1}$$

*where $D$ is a diagonal matrix with positive entries, $A$ encodes a DAG $G$ and $\epsilon \sim N(0, \mathrm{Id})$.*

For an illustration of the model, see Fig. 1a. It is convenient to encode the coefficients of linear SCMs through the matrix $B = D^{-1/2}(\mathrm{Id} - A)$. Then $Z = B^{-1}\epsilon$ and both $D$ and $A$ can be recovered from $B$, indeed the diagonal entries of $B$ agree with the entries of $D^{-1/2}$. Note that we can and will always assume that the entries of $D$ are positive since $\epsilon \overset{\mathcal{D}}{=} -\epsilon$.

Assuming that the latent variables follow a linear Gaussian SCM is certainly restrictive but nevertheless a reasonable approximation of the underlying data generating process in many settings. The Gaussian prior assumption is also standard in latent variable modeling and enables efficient inference for downstream tasks, among other advantages. Importantly, under the above assumptions, our model has infinite modeling capacity [53, 86, 33], so they're able to model complex datasets such as images and there's no loss in representational power.

The goal of representation learning is to learn the non-linear mixing $f$ and the high-level latent distribution $Z$ from observational data $X$. In causal representation learning, we wish to go a step further and model $Z$ as well by learning the parameters $B$ which then lets us easily recover $A, D$ and the causal graph $G$. For general non-linear mixing, this model is not identifiable and hence we cannot hope to recover $Z$, but in this work we use additional interventional data, similar to the setups in recent works [2, 84, 88].

**Assumption 3** (Single node interventions). *We consider interventional distributions $Z^{(i)}$ for $i \in I$ which are single node interventions of the latent distribution $Z$, i.e., for each intervention $i$ there is a target node $t_i$ and we change the assigning equation of $Z_{t_i}^{(i)}$ to*

$$Z_{t_i}^{(i)} = (A^{(i)} Z^{(i)})_{t_i} + (D^{(i)})_{t_i,t_i}^{1/2}(\epsilon_{t_i} + \eta^{(i)}) \tag{2}$$

*while leaving all other equations unchanged. We assume that $A_{t_i,k}^{(i)} \neq 0$ only if $k \rightarrow t_i$ in $G$, i.e., no parents are added and $\eta^{(i)}$ denotes a shift of the mean.*

For an illustration, see Fig. 1b. An intervention on node $t_i$ has no effect on the non-descendants of $t_i$, but will have a downstream effect on the descendants of $t_i$. In particular, $A$ is modified so that the weight of the edge $k \rightarrow t_i$ could potentially be changed if it exists already but no new incoming edges to $t_i$ may be added. Also, the noise variable $\epsilon_{t_i}$ is also allowed to be modified via a scale intervention as well as a shift intervention. Note that [84, 2] do not allow shift interventions, whereas we do.

Again, this can be written concisely as $Z^{(i)} = (B^{(i)})^{-1}(\epsilon + \eta^{(i)}e_{t_i})$ where $B^{(i)} = (D^{(i)})^{-1/2}(\mathrm{Id} - A^{(i)})$. Let $r^{(i)}$ denote the row $t_i$ of $B^{(i)}$, then we can write $B^{(i)} = B - e_{t_i}(B^\top e_{t_i} - r^{(i)})^\top$. We assume that interventions are non-trivial, i.e., $Z^{(i)} \overset{\mathcal{D}}{\neq} Z$. In our model, we observe interventional distributions $X^{(i)} = f(Z^{(i)})$ for various interventions $i \in I$.

**Definition 1** (Intervention Types). *We call an intervention perfect (or stochastic hard) if $A_{t_i,\cdot}^{(i)} = 0$, i.e., we remove all connections to former parents and potentially change variance and mean of the noise variable. Note that for nodes without parents, either $D_{t_i t_i}^{(i)} \neq D_{t_i t_i}$ or $\eta^{(i)} \neq 0$ so that the intervention is non-trivial. We call an intervention a pure noise intervention if $A_{t_i,\cdot}^{(i)} = A_{t_i,\cdot}$ and $A_{t_i,\cdot} \neq 0$ (i.e., node $t_i$ has at least one parent). In other words, a pure noise intervention targets a*

*node with parents by changing only the noise distribution through a change of variance (encoded in D) or a mean shift (encoded in $\eta$).*

We call a single-node intervention imperfect if it is not perfect (some prior works call it a soft intervention). Note that perfect interventions are never of pure noise type because they necessarily change the relation to the parents. For an illustration of perfect and imperfect interventions, see Fig. 1c, Fig. 1b respectively. Finally, we require an exhaustive set of interventions.

**Assumption 4** ((Coverage of Interventions)). *All nodes are intervened upon by at least one intervention, i.e., $\{t_i : i \in I\} = [d]$*

This assumption was also made in the prior works [84, 2, 88]. When the interventions don't cover all the nodes, we have non-identifiability as described in Section 4 and Appendix D. We also extensively discuss that none of our other assumptions can simply be dropped in these sections.

**Remark 1.**
  - *Our theory also readily extends to noisy observations, i.e., $X = f(Z) + \nu$ where $\nu$ is independent noise. In this case, we first denoise via a standard deconvolution argument [36, 42] and then apply our theory.*

  - *Similar to the results in [84] we can assume completely unknown intervention targets, i.e., there might be multiple interventions targeting the same node, and we do not need to know the partition. We only require coverage of all nodes. In contrast to their work we assume that we know which dataset corresponds to the observational distribution, but we expect that this restriction can be removed.*

To simplify the notation, it is convenient to use $B^{(0)} = B$, $\eta^{(0)} = 0$ for the observational distribution. We also define $\bar{I} = I \cup \{0\}$. Then, all information about the latent variable distributions $Z^{(i)}$ and observed distributions $X^{(i)}$ are contained in $((B^{(i)}, \eta^{(i)}, t_i)_{i \in \bar{I}}, f)$.

## 4   Main Results

We can now state our main results for the setting introduced above.

**Theorem 1.** *Suppose we are given distributions $X^{(i)}$ generated using a model $((B^{(i)}, \eta^{(i)}, t_i)_{i \in \bar{I}}, f)$ such that Assumptions 1-4 hold and such that all interventions $i$ are perfect. Then the model is identifiable up to permutation and scaling, i.e., for any model $((\widetilde{B}^{(i)}, \widetilde{\eta}^{(i)}, \widetilde{t}_i)_{i \in \bar{I}}, \widetilde{f})$ that generates the same data, the latent dimension $d$ agrees and there is a permutation $\omega \in S_d$ (and associated permutation matrix $P_\omega$) and an invertible pointwise scaling matrix $\Lambda \in \mathrm{Diag}(d)$ such that*

$$\widetilde{t}_i = \omega(t_i), \quad \widetilde{B}^{(i)} = P_\omega^\top B^{(i)} \Lambda^{-1} P_\omega, \quad \widetilde{f} = f \circ \Lambda^{-1} P_\omega, \quad \widetilde{\eta}^{(i)} = \eta^{(i)}. \tag{3}$$

*This in particular implies that*

$$\widetilde{Z}^{(i)} \stackrel{\mathcal{D}}{=} P_\omega^\top \Lambda Z^{(i)} \tag{4}$$

*and we can identify the causal graph $G$ up to permutation of the labels.*

This result says that for the interventional model as described in the previous section, we can identify the non-linear map $f$, the intervention targets $t_i$, the parameter matrices $B, D, A$ up to permutations $P_\omega$ and diagonal scaling $\Lambda$. Moreover, we can recover the shifts $\eta^{(i)}$ exactly and also the underlying causal graph $G$ up to permutations.

**Remark 2.** *Identifiability and recovery up to permutation and scaling is the best possible for our setting. This is because the latent variables $Z$ are not actually observed, which means we cannot (and in fact don't need to) resolve permutation and scaling ambiguity without further information about $Z$. See [84, Proposition 1] for the short proof.*

When we drop the assumption that the interventions are perfect, we can still obtain a weaker identifiability result. Define $\prec_G$ to be the minimal partial order on $[d]$ such that $i \prec_G j$ if $(i, j)$ is an edge in $G$, i.e., $i \prec_G j$ if and only if $i$ is an ancestor of $j$ in $G$. Note that any topological ordering of $G$ is compatible with the partial order $\prec_G$. Then, our next result shows that under imperfect interventions, we can still recover the partial order $\prec_G$.

**Theorem 2.** *Suppose we are given the distributions $X^{(i)}$ generated using a model $((B^{(i)}, \eta^{(i)}, t_i)_{i \in \bar{I}}, f)$ with causal graph $G$ such that the Assumptions 1-4 hold and none of the interventions is a pure noise intervention. Then for any other model $((\widetilde{B}^{(i)}, \widetilde{\eta}^{(i)}, \widetilde{t}_i)_{i \in \bar{I}}, \widetilde{f})$ with causal graph $\widetilde{G}$ generating the same observations the latent dimension $d$ agrees and there is a permutation $\omega \in S_d$ such that $\widetilde{t}_i = \omega(t_i)$ and $i \prec_{\widetilde{G}} j$ iff $\omega(i) \prec_G \omega(j)$, i.e., $\prec_G$ can be identified up to a permutation of the labels.*

**Remark 3.** *We emphasize that neither the full graph nor the coefficients $B^{(i)}$ or the latent variables $Z_i$ are identifiable in this setting, even for linear mixing functions.*

Theorem 1 and Theorem 2 generalize the main results of [84] which assume linear $f$ while we allow for non-linear $f$. The key new ingredient of our work is the following theorem, which shows identifiability of $f$ up to linear maps.

**Theorem 3.** *Assume that $X^{(i)}$ is generated according to a model $((B^{(i)}, \eta^{(i)}, t_i)_{i \in \bar{I}}, f)$ such that the Assumptions 1-4 hold. Then we have identifiability up to linear transformations, i.e., if $((\widetilde{B}^{(i)}, \widetilde{\eta}^{(i)}, \widetilde{t}_i), \widetilde{f})$ generates the same observed distributions $\widetilde{f}(\widetilde{Z}^{(i)}) \overset{\mathcal{D}}{=} X^{(i)}$, then their latent dimensions $d$ agree and there is an invertible linear map $T$ such that*

$$\widetilde{f} = f \circ T^{-1}, \quad \widetilde{Z}^{(i)} = T Z^{(i)}. \tag{5}$$

The proof of this theorem is deferred to Appendix A. In Appendix B we then review the results of [84] and show how their results can be extended to obtain our main results, Theorem 1 and Theorem 2. Now we provide some intuition for the proofs of the main theorems.

**Proof intuition** The recent work [84] studies the special case when $f$ is linear, and the proof is linear algebraic. In particular, they consider row spans of the precision matrices of $X^{(i)}$, project them to certain linear subspaces and use those subspaces to construct a generalized RQ decomposition of the pseudoinverse of the linear mixing matrix $f$. However, once we are in the setting of non-linear $f$, such an approach is not feasible because we can no longer reason about row spans of the precision matrices of $X$, which have been transformed non-linearly thereby losing all linear algebraic structure. Instead, we take a statistical approach and look at the log densities of the $X^{(i)}$. By choosing a Gaussian prior, the log-odds $\ln p_X^{(i)}(x) - \ln p_X^{(0)}(x)$ of $X^{(i)}$ with respect to $X^{(0)}$ can be written as a quadratic form of difference of precision matrices, evaluated at non-linear functions of $x$. For simplicity of this exposition, ignore terms arising from shift interventions and determinants of covariance matrices. Then, the log-odds looks like $\theta(x) = f^{-1}(x)^\top (\Theta^{(i)} - \Theta^{(0)}) f^{-1}(x)$, where $\Theta^{(i)}$ is the precision matrix of $Z^{(i)}$.

At this stage, we again shift our viewpoint to geometric and observe that $\Theta^{(i)} - \Theta^{(0)}$ has a certain structure. In particular, for single-node interventions, it has rank at most 2 and for source node targets, it has rank 1. This implies that the level set manifolds of the quadratic forms $\theta(x)$ also have a certain geometric structure in them, i.e., the DAG leaves a geometric signature on the data likelihood. We exploit this carefully and proceed by induction on the topological ordering until we end up showing that $f$ can be identified up to a linear transformation, which is our main Theorem 3. Here, the presence of shift interventions adds additional complexities, and we have to generalize all of our intermediate lemmas to handle these. Once we identify $f$ up to a linear transformation, we can apply the results of [84] to conclude Theorems 1 and 2.

**On the optimality and limitations of the assumptions** We make a few brief remarks on our assumptions, deferring to Appendix D a full discussion and the technical construction of several illustrative counterexamples.

1. *Number of interventions:* For our main results, Theorems 1 and 2, we assume that there are at least $d$ interventions (Assumption 4). This cannot be weakened even for linear mixing functions [84]. In addition, we also show in Fact 1 that for the linear identifiability proved in Theorem 3, $d-2$ interventions are not sufficient. Thus, the required number of interventions is tight in Theorems 1, 2 and is tight up to at most one intervention in Theorem 3.

2. *Intervention type:* In the setting of imperfect interventions, the weaker identifiability guarantees in Theorem 2 cannot be improved even when the mixing is linear [84, Appendix C]. We also show in Fact 2 that if we drop the condition that interventions are not pure noise interventions, we have non-identifiability. Concretely, we show that when all interventions

are of pure noise type, any causal graph is compatible with the observations. Finally, in contrast to the special case of linear mixing, we show in Lemma 8 that we need to exclude non-stochastic hard interventions (i.e., $\mathrm{do}(Z_i = z_i)$) for identifiability up to linear maps.

3. *Distributional assumptions:* We assume Gaussian latent variables, while allowing for very flexible mixing $f$. This model has universal approximation guarantees and is moreover used ubiquitously in practice. While the result can potentially be extended to more general latent distributions (e.g., exponential families), we additionally show in Lemma 9 that the result is not true when making no assumption on the distribution of $\epsilon$.

## 5   Experimental methodology

In this section, we explain our experimental methodology and the theoretical underpinning of our approach. Our main experiments for interventional causal representation learning focus on a method based on contrastive learning. We train a deep neural network to learn to distinguish observational samples $x \sim X^{(0)}$ from interventional samples $x \sim X^{(i)}$. Additionally, we design the last layer of the model to model the log-likelihood of a linear Gaussian SCM. Due to representational flexibility of deep neural networks, we will in principle learn the Bayes optimal classifier after optimal training, which we show is related to the underlying causal model parameters. Accordingly, with careful design of the last layer parametric form, we indirectly learn the parameters of the underlying causal model. Similar methods have been used for time-series data or multimodal data [27, 32] but to the best of our knowledge, the contrastive learning approach to interventional learning is novel.

Denote the probability density of $x \sim X^{(i)}$ (resp. $z \sim Z^{(i)}$) by $p_X^{(i)}(x)$ (resp. $p_Z^{(i)}(z)$). The next lemma describes the log-odds of a sample $x$ coming from the interventional distribution $X^{(i)}$ as opposed to the observational distribution $X^{(0)}$. We focus on the identifiable case (see Theorem 1) and therefore only consider perfect interventions. As per the notation in Section 3 and Appendix A.1, let $s^{(i)}$ denote the row $t_i$ of $B^{(0)}$ and let $\eta^{(i)}, \lambda_i$ denote the magnitude of the shift and scale intervention respectively.

**Lemma 1.** *When we have perfect interventions, the log-odds of a sample $x$ coming from $X^{(i)}$ over coming from $X^{(0)}$ is given by*

$$\ln p_X^{(i)}(x) - \ln p_X^{(0)}(x) = c_i - \frac{1}{2}\lambda_i^2(((f^{-1}(x))_{t_i})^2 + \eta^{(i)}\lambda_i \cdot (f^{-1}(x))_{t_i}) + \frac{1}{2}\langle f^{-1}(x), s^{(i)}\rangle^2 \quad (6)$$

*for a constant $c_i$ independent of $x$.*

The proof is deferred to Appendix F. The form of the log-odds suggests considering the following functions

$$g_i(x, \alpha_i, \beta_i, \gamma_i, w^{(i)}, \theta) = \alpha_i - \beta_i h_{t_i}^2(x, \theta) + \gamma_i h_{t_i}(x, \theta) + \langle h(x, \theta), w^{(i)}\rangle^2 \quad (7)$$

where $h(\cdot, \theta)$ denotes a neural net parametrized by $\theta$, parameters $w^{(i)}$ are the rows of a matrix $W$ and $\alpha_i, \beta_i, \gamma_i$ are learnable parameters. Note that the ground truth parameters minimize the following cross entropy loss

$$\mathcal{L}_{\mathrm{CE}}^{(i)} = \mathbb{E}_{j \sim \mathcal{U}(\{0,i\})}\mathbb{E}_{x \sim X^{(j)}}\mathrm{CE}(\mathbf{1}_{j=i}, g_i(x)) = -\mathbb{E}_{j \sim \mathcal{U}(\{0,i\})}\mathbb{E}_{x \sim X^{(j)}}\ln\left(\frac{e^{\mathbf{1}_{j=i}g_i(x)}}{e^{g_i(x)} + 1}\right). \quad (8)$$

Note that (compare (6) and (7)) $W$ should learn $B = D^{-1/2}(\mathrm{Id} - A)$ thus its off-diagonal entries should form a DAG. To enforce this we add the NOTEARS regularizer [102] given by $\mathcal{R}_{NOTEARS}(W) = \mathrm{tr}\exp(W_0 \circ W_0) - d$ (see Appendix F.3) where $W_0$ equals $W$ with the main diagonal zeroed out. We also promote sparsity by adding the $l_1$ regularization term $\mathcal{R}_{REG}(W) = \|W_0\|_1$. Thus, the total loss is given by

$$\mathcal{L}(\alpha, \beta, \gamma, W, \theta) = \sum_{i \in I}\mathcal{L}_{\mathrm{CE}}^{(i)} + \tau_1\mathcal{R}_{NOTEARS}(W) + \tau_2\mathcal{R}_{REG}(W) \quad (9)$$

for hyperparameters $\tau_1$ and $\tau_2$. Our identifiability result Theorem 1 implies that when we assume that the neural network has infinite capacity, the loss in (9) is minimized, $\tau_1$ is large and $\tau_2$ small, and we learn Gaussian latent variables $h(X^{(i)}, \theta)$, then we recover the ground truth latent variables up to the

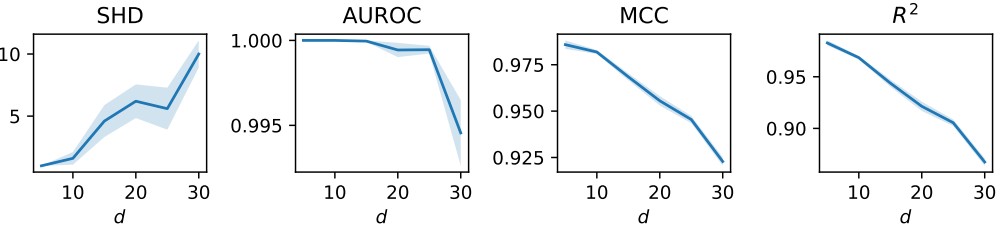

Figure 2: Dependence of performance metrics for $ER(d, 2)$ graphs with $d' = 100$ and nonlinear mixing $f$ on the dimension $d$.

tolerable ambiguities of labeling and scale, i.e., $h$ recovers $f^{-1}$ and $W$ recovers $B$ up to permutation and scale. Thus, we estimate the latent variables using $\hat{Z} = h(X, \theta)$ and estimate the DAG using $W_0$. Full details of the practical implementation of our approach are given in Appendix H.

Our experimental setup is similar to [84, 2], we consider $d$ interventions with different targets (our theory holds in full generality) and therefore, we can arbitrarily assign $t_i = i$ based on the intervention index $i$ which removes the permutation ambiguity. We focus on non-zero shifts because the cross entropy loss together with the quadratic expression for the log-odds results in a non-convex output layer which makes it hard to find the global loss minimizer, as we will describe in more detail in Appendix G.1. We emphasize that this is no contradiction to the theoretical results stated above. Even when there is no shift intervention the latent variables are identifiable, but our algorithm often fails to find the global minimizer of the loss (9) due to the non-convex loss landscape. For the sake of exposition and to set the stage for future work, we also briefly describe an approach via Variational Autoencoders (VAE). VAEs have been widely used in causal representation learning and while feasible in interventional settings, they are accompanied by certain difficulties, which we detail and suggest how to overcome in Appendix F.4.

## 6 Experiments

In this section, we implement our approach on synthetic data and image data. Complete details (architectures, hyperparameters) are deferred to Appendix H. Additional experiments investigating the effect of varsortability [66] and the noise distribution can be found in Appendix G.

**Data generation** For all our experiments we use Erdös-Rényi graphs, i.e., we add each undirected edge with equal probability $p$ to the graph and then finally orient them according to some random order of the nodes. We write $\mathrm{ER}(d, k)$ for the Erdös-Rényi graph distribution on $d$ nodes with $kd$ expected edges. For a given graph $G$ we then sample edge weights from $\mathcal{U}(\pm[0.25, 1.0])$ and a scale matrix $D$. For simplicity we assume that we have $n$ samples from each environment $i \in \bar{I}$. We only consider the setting where each node is intervened upon once and thus the latent dimension is also known. We consider three types of mixing functions. First, we consider linear mixing functions where we sample all matrix entries i.i.d. from a Gaussian distribution. Then, we consider non-linear mixing functions that are parametrized by MLPs with three hidden layers which are randomly initialized, and have Leaky ReLU activations. Finally, we consider image data as described in [2]. Pairs of latent variables $(z_{2i+1}, z_{2i+2})$ describe the coordinates of balls in an image and the non-linearity $f$ is the rendering of the image. The image generation is based on pygame [77]. A sample image can be found in Figure 3.

**Evaluation Metrics** We evaluate how well we can recover the ground truth latent variables and the underlying DAG. For the recovery of the ground truth latents we use the Mean Correlation Coefficient (MCC) [37, Appendix A.2]. For the evaluation of the learned graph structure we use the Structural Hamming Distance (SHD) where we follow the convention that we count the number of edge differences of the directed graphs (i.e., an edge with wrong orientation counts as two errors). Since the scale of the variables is not fixed the selection of the edge selection threshold is slightly delicate. Thus, we use the heuristic where we match the number of selected edges to the expected number

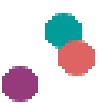

Figure 3: Sample image with 3 balls.

of edges. As a metric that is independent of this thresholding procedure, we also report the Area Under the Receiver Operating Curve (AUROC) for the edge selection.

**Methods** We implement our contrastive algorithm as explained in Section F where we use an MLP (with Leaky ReLU nonlinearities) for the function $h$ for the linear and nonlinear mixing functions and a very small convolutional network for the image dataset, which are termed "Contrastive". For linear mixing function we also consider a version of the contrastive algorithm where $h$ is a linear function, termed "Contrastive Linear". As baselines, we consider a variational autoencoder with latent dimension $d$ and also the algorithm for linear disentanglement introduced in [84]. Since the variational autoencoder does not output a causal graph structure we report the result for the empty graph here which serves as a baseline.

**Results for linear $f$** First, for the sake of comparison, we replicate exactly the setting from [84], i.e., we consider initial noise variances sampled uniformly from $[2, 4]$ and we consider perfect interventions where the new variance is sampled uniformly from $[6, 8]$. We set $d = 5$, $d' = 10$, $k = 3/2$, and $n = 50000$. Results can be found in Table 1. The linear contrastive method identifies the ground truth latent variables up to scaling. The failure of the nonlinear method to recover the ground truth latents will be explained and further analyzed in Appendix G.1. Note that the nonlinear contrastive and the linear contrastive method recover the underlying graph better than the baseline.

Table 1: Results for linear $f$ with $d = 5, d' = 10, k = 3/2, n = 50000$.

| Method | SHD ↓ | AUROC ↑ | MCC ↑ | $R^2$ ↑ |
|---|---|---|---|---|
| **Contrastive** | $4.6 \pm 0.5$ | $0.84 \pm 0.02$ | $0.05 \pm 0.02$ | $0.02 \pm 0.00$ |
| **Contrastive Linear** | $5.4 \pm 1.6$ | $0.80 \pm 0.07$ | $0.90 \pm 0.03$ | $1.00 \pm 0.00$ |
| Linear baseline | $7.0 \pm 0.5$ | $0.64 \pm 0.05$ | $0.83 \pm 0.04$ | $1.00 \pm 0.00$ |

**Results for nonlinear $f$** We sample all variances from $\mathcal{U}([1, 2])$ (initial variance and resampling for the perfect interventions) and the shift parameters $\eta^{(i)}$ of the interventions from $\mathcal{U}([1, 2])$. Results can be found in Table 2. We find that our contrastive method can recover the ground truth latents and the causal structure almost perfectly, while the baseline for linear disentanglement cannot recover the graph or the latent variables (which is not surprising as the mixing is highly nonlinear). Also, training a vanilla VAE does not recover the latent variables up to a linear map as indicated by the $R^2$ scores. In Figure 2 we illustrate the dependence on the dimension $d$ of our algorithm in this setting.

Table 2: Results for nonlinear synthetic data with $n = 10000$.

| Setting | Method | SHD ↓ | AUROC ↑ | MCC ↑ | $R^2$ ↑ |
|---|---|---|---|---|---|
| | **Contrastive** | $1.8 \pm 0.5$ | $0.97 \pm 0.01$ | $0.97 \pm 0.00$ | $0.96 \pm 0.00$ |
| ER(5, 2), $d' = 20$ | VAE | $10.0 \pm 0.0$ | $0.50 \pm 0.00$ | $0.48 \pm 0.03$ | $0.57 \pm 0.07$ |
| | Linear baseline | $10.6 \pm 1.9$ | $0.48 \pm 0.11$ | $0.32 \pm 0.03$ | $0.34 \pm 0.06$ |
| | **Contrastive** | $1.0 \pm 0.0$ | $1.00 \pm 0.00$ | $0.99 \pm 0.00$ | $0.98 \pm 0.00$ |
| ER(5, 2), $d' = 100$ | VAE | $10.0 \pm 0.0$ | $0.50 \pm 0.00$ | $0.59 \pm 0.02$ | $0.68 \pm 0.04$ |
| | Linear baseline | $3.4 \pm 1.2$ | $0.85 \pm 0.07$ | $0.18 \pm 0.04$ | $0.11 \pm 0.04$ |
| | **Contrastive** | $3.6 \pm 1.3$ | $0.98 \pm 0.01$ | $0.93 \pm 0.00$ | $0.87 \pm 0.01$ |
| ER(10, 2), $d' = 20$ | VAE | $18.6 \pm 0.9$ | $0.50 \pm 0.00$ | $0.59 \pm 0.02$ | $0.72 \pm 0.02$ |
| | Linear baseline | $29.6 \pm 2.5$ | $0.49 \pm 0.02$ | $0.44 \pm 0.02$ | $0.51 \pm 0.02$ |
| | **Contrastive** | $1.6 \pm 0.5$ | $1.00 \pm 0.00$ | $0.98 \pm 0.00$ | $0.97 \pm 0.00$ |
| ER(10, 2), $d' = 100$ | VAE | $18.6 \pm 0.9$ | $0.50 \pm 0.00$ | $0.62 \pm 0.02$ | $0.78 \pm 0.01$ |
| | Linear baseline | $28.4 \pm 2.1$ | $0.51 \pm 0.04$ | $0.17 \pm 0.03$ | $0.13 \pm 0.03$ |

**Results for image data** Finally, we report the results for image data. Here, we generate the graph as before and consider variances sampled from $\sigma^2 \sim \mathcal{U}([0.01, 0.02])$ and shifts $\eta^{(i)}$ from $\mathcal{U}([0.1, 0.2])$ (i.e., the shifts are still of order $\sigma$). We exclude samples where one of the balls is not contained in the image which generates a slight model misspecification compared to our theory. Again we find that we recover the latent graph and the latent variables as detailed in Table 3.

Table 3: Results for image data with ER($d$, 2) graphs with $d = 2 \cdot$ #balls and $n_{\text{int}} = 25000$ (per environment), $n_{\text{obs}} = n_{\text{int}} \cdot d$.

| # Balls | Method | SHD $\downarrow$ | AUROC $\uparrow$ | MCC $\uparrow$ | $R^2 \uparrow$ |
|---|---|---|---|---|---|
| 2 | Contrastive Learning | $1.4 \pm 0.4$ | $0.95 \pm 0.03$ | $0.87 \pm 0.03$ | $0.84 \pm 0.03$ |
| | VAE | $6.0 \pm 0.0$ | $0.50 \pm 0.00$ | $0.19 \pm 0.06$ | $0.16 \pm 0.08$ |
| 5 | Contrastive Learning | $2.0 \pm 0.3$ | $1.00 \pm 0.00$ | $0.94 \pm 0.01$ | $0.91 \pm 0.01$ |
| | VAE | $18.6 \pm 0.9$ | $0.50 \pm 0.00$ | $0.31 \pm 0.02$ | $0.36 \pm 0.03$ |
| 10 | Contrastive Learning | $11.0 \pm 3.3$ | $0.98 \pm 0.02$ | $0.89 \pm 0.01$ | $0.83 \pm 0.01$ |
| | VAE | $37.2 \pm 3.1$ | $0.50 \pm 0.00$ | $0.22 \pm 0.01$ | $0.33 \pm 0.02$ |

## 7   Conclusion

In this work, we extend several prior works and show identifiability for a widely used class of linear latent variable models with non-linear mixing, from interventional data with unknown targets. Counterexamples show that our assumptions are tight in this setting and could only potentially be relaxed under other additional assumptions. We leave to future work to extend our results for other classes of priors, such as non-parametric distribution families, and also to study sample complexity and robustness of our results. We also proposed a contrastive approach to learn such models in practice and showed that it can recover the latent structure in various settings. Finally, we highlight that the results of our experiments are very promising and it would be interesting to scale up our algorithms to large-scale datasets such as [49].

## Acknowledgments

We thank anonymous reviewers for useful comments and suggestions. We acknowledge the support of AFRL and DARPA via FA8750-23-2-1015, ONR via N00014-23-1-2368, and NSF via IIS-1909816, IIS-1955532. We also acknowledge the support of JPMorgan Chase & Co. AI Research. This work was also supported by the Tübingen AI Center.

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

# A  Proof of Identifiability up to Linear Maps

In this section we give a full proof of Theorem 3. We try to make this essential self-contained. To make this easy to read, we first introduce and recall the required notation in Section A.1, then we prove two important key relations required for the proof in Section A.2. Finally, we prove first a simpler version of Theorem 3 with more assumptions in Section A.3 and then give the full proof of Theorem 3 in Section A.4.

## A.1  General Notation

Let us introduce some notation for our identifiability proofs. For a full list of assumptions we refer to the main text, here we just recall the relevant notation concisely. We will assume that there are two representations that generate the same observations, i.e., we assume that there are two sets of Gaussian SCMs

$$Z^{(i)} = A^{(i)} Z^{(i)} + (D^{(i)})^{\frac{1}{2}} (\epsilon + \eta^{(i)} e_{t_i}) \tag{10}$$

$$\widetilde{Z}^{(i)} = \widetilde{A}^{(i)} \widetilde{Z}^{(i)} + (\widetilde{D}^{(i)})^{\frac{1}{2}} (\widetilde{\epsilon} + \widetilde{\eta}^{(i)} e_{\widetilde{t}_i}) \tag{11}$$

where $\epsilon, \widetilde{\epsilon} \sim N(0, \mathrm{Id})$ and $i = 0$ corresponds to the observational distribution while $i > 0$ correspond to interventional settings where we intervene on a single node $t_i$ (or $\widetilde{t}_i$) by adding a shift and changing the relation to its parents (without adding new parents). Moreover, there are two functions $f$, $\widetilde{f}$ such that $X^{(i)} \overset{\mathcal{D}}{=} f(Z^{(i)})$ and $X^{(i)} \overset{\mathcal{D}}{=} \widetilde{f}(\widetilde{Z}^{(i)})$. It is convenient to define

$$B^{(i)} = (D^{(i)})^{-\frac{1}{2}} (\mathrm{Id} - A^{(i)}) \tag{12}$$

$$\mu^{(i)} = \eta^{(i)} (B^{(i)})^{-1} e_{t_i} \tag{13}$$

so that $Z^{(i)} = (B^{(i)})^{-1} \epsilon + \mu^{(i)}$ with $\epsilon \sim N(0, \mathrm{Id})$. Note that all model information can be collected in $((B^{(i)}, \eta^{(i)}, t_i), f)$.

As in the main text, it is convenient to use the shorthand $B = B^{(0)}$, $A = A^{(0)}$.

We denote the row $t_i$ of $B^{(i)}$ by $r^{(i)}$, i.e.,

$$r^{(i)} = (B^{(i)})^\top e_{t_i}. \tag{14}$$

Note that for perfect interventions where we cut the relation to all parents of node $t_i$ we have $r^{(i)} = \lambda_i e_{t_i}$ for some $\lambda_i \neq 0$. Similarly, for the observational distribution we refer to the row $t_i$ of $B^{(0)}$ by $s^{(i)}$, i.e.,

$$s^{(i)} = (B^{(0)})^\top e_{t_i}. \tag{15}$$

We thus find

$$B^{(0)} - B^{(i)} = e_{t_i} (e_{t_i}^\top B^{(0)} - e_{t_i}^\top B^{(i)}) = e_{t_i} (s^{(i)} - r^{(i)})^\top. \tag{16}$$

We also consider the covariance matrix $\Sigma^{(i)}$ and the precision matrix $\Theta^{(i)}$ of $Z^{(i)}$ which are given by

$$\Theta^{(i)} = (B^{(i)})^\top B^{(i)}, \quad \Sigma^{(i)} = (\Theta^{(i)})^{-1} \tag{17}$$

Indeed,

$$\Sigma^{(i)} = \mathbb{E}[(B^{(i)})^{-1} \epsilon \epsilon^\top (B^{(i)})^{-\top}] = (B^{(i)})^{-1} \mathbb{E}[\epsilon \epsilon^\top] (B^{(i)})^{-\top} = (B^{(i)})^{-1} (B^{(i)})^{-\top}$$

Finally, the mean $\mu^{(i)}$ of $Z^{(i)}$ given by

$$\mu^{(i)} = \mathbb{E} Z^{(i)} = \eta^{(i)} (B^{(i)})^{-1} e_{t_i}. \tag{18}$$

We also define similar quantities for $\widetilde{Z}$. To simplify the notation, we will frequently use the shorthand $v \otimes w = v \cdot w^\top \in \mathbb{R}^{d \times d}$ for $v, w \in \mathbb{R}^d$.

### A.2 Two auxiliary lemmas

The key to our identifiability results is the relation shown in the following lemma.

**Lemma 2.** *Assume that there are two latent variable models $((B^{(i)}, \eta^{(i)}, t_i)_{i \in I}, f)$ and $((\widetilde{B}^{(i)}, \widetilde{\eta}^{(i)}, \widetilde{t}_i)_{i \in I}, \widetilde{f})$ as defined above such that $f(Z^{(i)}) \overset{\mathcal{D}}{=} \widetilde{f}(\widetilde{Z}^{(i)})$. Then their latent dimension $d$ agree and $h = \widetilde{f}^{-1} \circ f$ exists and is a diffeomorphism, i.e., bijective and differentiable with differentiable inverse. Moreover, it satisfies the relation*

$$
\frac{1}{2} z^\top (\Theta^{(i)} - \Theta^{(0)}) z - \eta^{(i)} (r^{(i)})^\top z
$$
$$
= \frac{1}{2} h(z)^\top (\widetilde{\Theta}^{(i)} - \widetilde{\Theta}^{(0)}) h(z) - \widetilde{\eta}^{(i)} (\widetilde{r}^{(i)})^\top h(z) + c^{(i)} \tag{19}
$$

*for all $z$ and $i$ and some constant $c^{(i)}$. If $\eta^{(i)} = \widetilde{\eta}^{(i)} = 0$ this simplifies to*

$$
\frac{1}{2} z^\top (\Theta^{(i)} - \Theta^{(0)}) z = \frac{1}{2} h(z)^\top (\widetilde{\Theta}^{(i)} - \widetilde{\Theta}^{(0)}) h(z) + c^{(i)}. \tag{20}
$$

*Proof.* Note first that $X^{(0)} \overset{\mathcal{D}}{=} f(Z^{(0)}$ implies since $f$ is an embedding by assumption that the dimension of the support of $X^{(0)}$ is the same as the latent dimension and this dimension is identifiable from the observations, e.g., by considering the tangent space. As $Z^{(0)}$ and $\widetilde{Z}^{(0)}$ have full support and $f$ and $\tilde{f}$ are by assumption embeddings, the equality $f(Z^{(0)}) \overset{\mathcal{D}}{=} \widetilde{f}(\widetilde{Z}^{(0)})$ implies that the two image manifolds agree and $h = \widetilde{f}^{-1} \circ f$ is a differentiable map. Moreover, we find that $h_* Z^{(i)} = \widetilde{Z}^{(i)}$ where $h_*$ denotes the pushforward map. The variables $Z^{(i)}$ are Gaussian with distribution $Z^{(i)} \sim N(\mu^{(i)}, \Sigma^{(i)})$. This implies that its density $p^{(i)}$ can be written as

$$
\ln(p^{(i)}(z)) = -\frac{n}{2} \ln(2\pi) - \frac{1}{2} \ln |\Sigma^{(i)}| - \frac{1}{2} (z - \mu^{(i)})^\top \Theta^{(i)} (z - \mu^{(i)}). \tag{21}
$$

A similar relation holds for $\widetilde{p}^{(i)}(z)$, the density of $\widetilde{Z}^{(i)}$. Finally, we have the relations

$$
p^{(i)}(z) = \widetilde{p}^{(i)}(h(z)) |\det J_h(z)| \tag{22}
$$

which is a consequence of $h_* Z^{(i)} = \widetilde{Z}^{(i)}$. Here $J_h$ denotes the Jacobian matrix of partial derivatives. Thus we conclude that

$$
-\frac{n}{2} \ln(2\pi) - \frac{1}{2} \ln |\Sigma^{(i)}| - \frac{1}{2} (z - \mu^{(i)})^\top \Theta^{(i)} (z - \mu^{(i)})
$$
$$
= -\frac{n}{2} \ln(2\pi) - \frac{1}{2} \ln |\widetilde{\Sigma}^{(i)}| - \frac{1}{2} (h(z) - \widetilde{\mu}^{(i)})^\top \widetilde{\Theta}^{(i)} (h(z) - \widetilde{\mu}^{(i)}) + \ln |\det J_h(z)|. \tag{23}
$$

Calling this relation $E^i$, we consider the difference $E^0 - E^i$. Then the determinant term cancels, and we obtain for some constant $c^{(i)}$

$$
\frac{1}{2} (z - \mu^{(i)})^\top \Theta^{(i)} (z - \mu^{(i)}) - \frac{1}{2} z^\top \Theta^{(0)} z
$$
$$
= \frac{1}{2} (h(z) - \widetilde{\mu}^{(i)})^\top \widetilde{\Theta}^{(i)} (h(z) - \widetilde{\mu}^{(i)}) - \frac{1}{2} h(z)^\top \widetilde{\Theta}^{(0)} h(z) + c'^{(i)}. \tag{24}
$$

Expanding the quadratic expression on the left-hand side we obtain

$$
\frac{1}{2} (z - \mu^{(i)})^\top \Theta^{(i)} (z - \mu^{(i)}) - \frac{1}{2} z^\top \Theta^{(0)} z = \frac{1}{2} z^\top (\Theta^{(i)} - \Theta) z - z^\top \Theta^{(i)} \mu^{(i)} + \frac{1}{2} (\mu^{(i)})^\top \Theta^{(i)} \mu^{(i)}. \tag{25}
$$

Finally we simplify the last term using (18)

$$
\Theta^{(i)} \mu^{(i)} = \eta^{(i)} (B^{(i)})^\top B^{(i)} (B^{(i)})^{-1} e_{t_i} = \eta^{(i)} (B^{(i)})^\top e_{t_i} = \eta^{(i)} r^{(i)} \tag{26}
$$

Apply the same simplification on the right hand side. Finally, we collect the constant terms independent of $z$ in $c^{(i)}$, to obtain

$$
\frac{1}{2} z^\top (\Theta^{(i)} - \Theta^{(0)}) z - \eta^{(i)} (r^{(i)})^\top z
$$
$$
= \frac{1}{2} h(z)^\top (\widetilde{\Theta}^{(i)} - \widetilde{\Theta}^{(0)}) h(z) - \widetilde{\eta}^{(i)} (\widetilde{r}^{(i)})^\top h(z) + c^{(i)} \tag{27}
$$

which is (19) as desired. $\qquad \square$

A second key ingredient for our identifiability results is that the specific structure of the interventions that each target a single node which implies that the difference of precision matrices $\Theta^{(i)} - \Theta^{(0)}$ has rank at most 2. This is also the key ingredient used (in a very different manner) in the proof of the main result in [84]. Here we highlight this simple result because we make use of it frequently.

**Lemma 3.** *The precision matrices satisfy the relation*

$$\Theta^{(i)} - \Theta^{(0)} = r^{(i)} \otimes r^{(i)} - s^{(i)} \otimes s^{(i)}. \tag{28}$$

*Proof.* Using (17) we find

$$
\begin{aligned}
\Theta^{(i)} - \Theta^{(0)} &= (B^{(i)})^\top B^{(i)} - B^\top B \\
&= (B^{(i)})^\top \left( \sum_{k=1}^{d} e_k e_k^\top \right) B^{(i)} - B^\top \left( \sum_{k=1}^{d} e_k e_k^\top \right) B \\
&= \sum_{k=1}^{d} (B^{(i)})^\top e_k ((B^{(i)})^\top e_k)^\top - \sum_{k=1}^{d} B^\top e_k (B^\top e_k)^\top \\
&= ((B^{(i)})^\top e_{t_i}) \otimes ((B^{(i)})^\top e_{t_i}) - (B^\top e_{t_i}) \otimes (B^\top e_{t_i}) \\
&= r^{(i)} \otimes r^{(i)} - s^{(i)} \otimes s^{(i)}.
\end{aligned} \tag{29}
$$

Here we used the fact that all rows except row $t_i$ of $B^{(i)}$ and $B^{(0)}$ agree as the intervention targets a single node, i.e., $(B^{(i)})^\top e_k = (B^{(0)})^\top e_k$ for $k \neq t_i$. $\qquad\square$

### A.3   Warm-up

To provide some intuition how identifiability arises, we first provide the proof for a simpler result with a much stronger assumption on the set of intervention targets and the type of interventions. But note importantly that we don't relax the assumption of non-linearity of $f$.

**Theorem 4.** *Let $f(Z^{(i)}) \stackrel{\mathcal{D}}{=} \widetilde{f}(\widetilde{Z}^{(i)})$ for two sets of parameters $((B^{(i)}, \eta^{(i)}, t_i)_{i \in I}, f)$ and $((\widetilde{B}^{(i)}, \widetilde{\eta}^{(i)}, \widetilde{t}_i)_{i \in I}, \widetilde{f})$ as in the above model. We assume that there are $2n$ perfect pairwise different interventions (i.e., $Z^{(i)} \stackrel{\mathcal{D}}{\neq} Z^{(i')}$ for $i \neq i'$), without shifts (i.e., $\eta^{(i)} = \widetilde{\eta}^{(i)} = 0$). Finally, we assume that there are 2 interventions for each node, and we know the intervention targets of the interventions, i.e., we assume that $t_i = \widetilde{t}_i$ for all $i$. Then, $Z^{(i)}$ and $\widetilde{Z}^{(i)}$ agree up to scaling, i.e., $Z^{(i)} = D\widetilde{Z}^{(i)}$ for a diagonal matrix $D \in \mathrm{Diag}(n)$.*

**Remark 4.** *Actually this result can be generalized to a setting where the latent SCM is non-parametric [92].*

*Proof.* We consider a node $k$ and two interventions $i_1$ and $i_2$ which target node $k = t_{i_1} = t_{i_2}$ (by assumption those interventions exist and $i_1$ and $i_2$ agree for both representations). Since we assumed that the interventions are perfect, we have in our notation that

$$r^{(i_1)} = \lambda_{i_1} e_k, \quad r^{(i_2)} = \lambda_{i_2} e_k \tag{30}$$

for two constants $\lambda_{i_1} \neq \lambda_{i_2} > 0$ and similarly for $\widetilde{r}^{(i_j)}$. Indeed, assuming positivity is not a restriction since $\epsilon$ follows a symmetric distribution (this is also contained in the parametrisation $(D^{(i)})^{\frac{1}{2}}$) and $\lambda_{i_1} \neq \lambda_{i_2}$ is necessary to ensure that the interventional distributions differ. Combining (20) from Lemma 2 with Lemma 3 we obtain the relation

$$
\begin{aligned}
z^\top \left( r^{(i_j)} \otimes r^{(i_j)} - s^{(i_j)} \otimes s^{(i_j)} \right) z &= z^\top (\Theta^{(i_j)} - \Theta^{(0)}) z \\
&= h(z)^\top (\widetilde{\Theta}^{(i_j)} - \widetilde{\Theta}^{(0)}) h(z) + 2c^{(i_j)} \\
&= h(z)^\top \left( \widetilde{r}^{(i_j)} \otimes \widetilde{r}^{(i_j)} - \widetilde{s}^{(i_j)} \otimes \widetilde{s}^{(i_j)} \right) h(z) + 2c^{(i_j)}.
\end{aligned} \tag{31}
$$

for $j = 1, 2$.

Note that since $k = t_{i_1} = t_{i_2}$, we have $s^{(i_1)} = s^{(i_2)} = B^\top e_k$ by definition. Thus, we subtract the relation in the last display for $i_1$ from the relation for $i_2$ and find using (30),

$$
\begin{aligned}
(\lambda_{i_1}^2 - \lambda_{i_2}^2)z_k^2 &= (z^\top r^{(i_1)})^2 - (z^\top r^{(i_2)})^2 \\
&= (h(z)^\top \widetilde{r}^{(i_1)})^2 - (h(z)^\top \widetilde{r}^{(i_2)})^2 + 2(c^{(i_1)} - c^{(i_2)}) \\
&= (\widetilde{\lambda}_{i_1}^2 - \widetilde{\lambda}_{i_2}^2)h_k(z)^2 + 2(c^{(i_1)} - c^{(i_2)}).
\end{aligned}
\tag{32}
$$

for all $z$. Since $\widetilde{\lambda}_{i_1} \neq \widetilde{\lambda}_{i_2} > 0$ we can divide and obtain

$$
\frac{\lambda_{i_1}^2 - \lambda_{i_2}^2}{\widetilde{\lambda}_{i_1}^2 - \widetilde{\lambda}_{i_2}^2}z_k^2 = h_k(z)^2 + 2\frac{c^{(i_1)} - c^{(i_2)}}{\widetilde{\lambda}_{i_1}^2 - \widetilde{\lambda}_{i_2}^2}.
\tag{33}
$$

Since $h$ is surjective, we have that $h_k$ is also surjective which implies that the range of the right hand side is $[2(c^{(i_1)} - c^{(i_2)})/(\widetilde{\lambda}_{i_1}^2 - \widetilde{\lambda}_{i_2}^2), \infty)$. The range of the left hand side is depending on the sign of the prefactor and is either $(-\infty, 0]$ or $[0, \infty)$. Since the ranges have to agree we conclude that $c^{(i_1)} = c^{(i_2)}$ and

$$
h_k(z) = \pm\sqrt{\frac{\lambda_{i_1}^2 - \lambda_{i_2}^2}{\widetilde{\lambda}_{i_1}^2 - \widetilde{\lambda}_{i_2}^2}}z_k.
\tag{34}
$$

Since $k$ was arbitrary, this ends the proof. $\qquad\square$

### A.4  Proof of identifiability up to linear maps

The proof of our main result relies on two essentially geometric lemmas and a slight extension of Lemma 2 which we discuss now. The first lemma alone is sufficient to handle the case where interventions change the scaling matrix $D$, while the second is required to handle interventions that change $B$ and the third allows us to consider pure shift interventions.

**Lemma 4.** *Let $g : \mathbb{R}^d \to \mathbb{R}$ be a continuous and surjective function, $Q$ a quadratic form on $\mathbb{R}^d$ and $\alpha \in \mathbb{R}$ a constant such that*

$$
g(z)^2 = z \cdot Qz + \alpha.
\tag{35}
$$

*Then $Q$ has rank 1 and $g$ is linear, i.e., $g(z) = v \cdot z$ for some $v \in \mathbb{R}^d$.*

*Proof.* Assume that $Q$ has the eigendecomposition $Q = \sum_{i=1}^r \lambda_i v_i v_i^\top$ where $r$ is the rank of $Q$ and $\lambda_i \neq 0$ and $v_i$ has unit norm. Clearly $g$ surjective implies $r > 0$. Since $g$ is surjective we find that the range of the left hand side is $[0, \infty)$ and therefore $z \cdot Qz \geq -\alpha$. This implies that $\lambda_i \geq 0$ for all $i$ and therefore, the range of the right hand side is $[\alpha, \infty)$ which implies $\alpha = 0$. Now we fix any $c > 0$ and consider $E_c = \{z \in \mathbb{R}^d \mid z \cdot Qz = c\}$. It is easy to see that $E_c$ is the product of an ellipsoid and $\mathbb{R}^{d-r}$ and thus is connected if $r \geq 2$. We find by (35) that $g(z) = \pm\sqrt{c}$ for $z \in E_c$ and if $g(z) \in \{-\sqrt{c}, \sqrt{c}\}$ then $z \in E_c$. By continuity of $g$ and since $E_c$ is connected this implies that $g(z) = \sqrt{c}$ for all $z \in E_c$ or $g(z) = -\sqrt{c}$ for all $z \in E_c$ contradicting the surjectivity of $g$. We conclude that $r = 1$, i.e., $Q = \lambda_1 v_1 v_1^\top = \tilde{v}_1 \tilde{v}_1^\top$ with $\tilde{v}_1 = \sqrt{\lambda_1} v_1$. Thus we find that

$$
g(z)^2 = |\tilde{v}_1 \cdot z|^2.
\tag{36}
$$

Since $g$ is continuous, this implies that $g(z) = \pm\tilde{v}_1 \cdot z$ or $g(z) = \pm|\tilde{v}_1 \cdot z|$. As only the first functions are surjective we conclude that $g(z) = \pm\tilde{v}_1 \cdot z$ and the claim follows with $w = \pm\tilde{v}_1$. $\qquad\square$

To handle shift interventions we need a slightly stronger version of the lemma above which contains an additional linear term.

**Corollary 1.** *Let $g : \mathbb{R}^d \to \mathbb{R}$ be a continuous and surjective function, $Q$ a quadratic form on $\mathbb{R}^d$, $w \in \mathbb{R}^d$ a vector, and $\alpha \in \mathbb{R}$ a constant such that*

$$
g(z)^2 = z \cdot Qz + w \cdot z + \alpha.
\tag{37}
$$

*Then $Q$ has rank 1 and $g$ is affine, i.e., $g(z) = v \cdot z + c$ for some $v \in \mathbb{R}^d$ and $c \in \mathbb{R}$.*

*Proof.* Assume as before that $Q$ has the eigendecomposition $Q = \sum_{i=1}^{r} \lambda_i v_i v_i^\top$ where $r$ is the rank of $Q$ and $\lambda_i \neq 0$ and $|v_i| = 1$. First, we show that $w \in \langle v_1, \ldots, v_r \rangle$ where $\langle . \rangle$ denotes the span. Indeed, suppose that this is not the case. Then we could find $z_0$ such that $z_0 \cdot w < 0$ and $z_0 v_i = 0$ for $1 \leq i \leq r$, and therefore $z_0 \cdot Q z_0 = 0$. Plugging $z = \lambda z_0$ for $\lambda \to \infty$ in (37) the right-hand side tends to $-\infty$ while the left-hand side is non-negative. This is a contradiction and therefore $w \in \langle v_1, \ldots, v_r \rangle$ holds. This implies that there is $w'$ such that $Q w' = w/2$ and thus

$$z \cdot Qz + w \cdot z + \alpha = (z + w') \cdot Q(z + w') + \alpha - w' \cdot Q w'. \tag{38}$$

Then we consider $\widetilde{g}(\widetilde{z}) = g(\widetilde{z} - w')$, $\widetilde{\alpha} = \alpha - w' \cdot Q w'$ which satisfy

$$\widetilde{g}^2(\widetilde{z}) = g^2(\widetilde{z} - w') \tag{39}$$
$$= (\widetilde{z} - w' + w') \cdot Q(\widetilde{z} - w' + w') + \alpha - w' \cdot Q w' \tag{40}$$
$$= \widetilde{z} \cdot Q\widetilde{z} + \widetilde{\alpha}. \tag{41}$$

Using Lemma 4 we conclude that $Q$ has rank 1 and $\widetilde{g}(\widetilde{z}) = v \cdot z$ for some $v$. But then $g(z) = \widetilde{g}(z + w') = z \cdot v + w' \cdot v$ which concludes the proof. $\qquad\square$

The second lemma is similar, but slightly simpler.

**Lemma 5.** *Assume that there is a quadratic form $Q$, a vector $0 \neq w \in \mathbb{R}^d$, $w' \in \mathbb{R}^d$, $\alpha \in \mathbb{R}$ and a continuous function $g : \mathbb{R}^d \to \mathbb{R}$ such that*

$$g(z)(w \cdot z) = z \cdot Qz + w' \cdot z + \alpha. \tag{42}$$

*Then $g(z) = v \cdot z + c$ for some $v \in \mathbb{R}^d$, $c \in \mathbb{R}$, i.e., $g$ is an affine function.*

*Proof.* By rescaling we can assume that $w$ has unit norm. Plugging in $z = 0$ we find that $\alpha = 0$. Denote by $\Pi_w$ the orthogonal projection on $w$, i.e., $\Pi_w v = (w \cdot v) w$ and by $\Pi_w^\perp$ the projection on the orthogonal complement of $w$, i.e., $z = \Pi_w z + \Pi_w^\perp z$ for all $z \in \mathbb{R}^d$. Then we find for all $z$

$$0 = g(\Pi_w^\perp z) w \cdot \Pi_w^\perp z = (\Pi_w^\perp z) \cdot Q\Pi_w^\perp z + (\Pi_w^\perp z) \cdot w'. \tag{43}$$

By scaling $z$ we find that both terms on the right hand side vanish and thus for all $z$

$$(\Pi_w^\perp z) \cdot Q\Pi_w^\perp z = 0, \tag{44}$$
$$(\Pi_w^\perp z) \cdot w' = 0. \tag{45}$$

This implies (using also the symmetry of $Q$)

$$\begin{aligned} g(z) w \cdot \Pi_w z &= g(z) w \cdot z \\ &= z \cdot Qz + w' \cdot z \\ &= (\Pi_w z + \Pi_w^\perp z) \cdot Q(\Pi_w z + \Pi_w^\perp z) + w' \cdot (\Pi_w z + \Pi_w^\perp z) \\ &= (\Pi_w z) \cdot Q\Pi_w z + 2(\Pi_w z) Q\Pi_w^\perp z + (\Pi_w^\perp z) \cdot Q\Pi_w^\perp z + \Pi_w z \cdot w' \\ &= (\Pi_w z) \cdot (Q(\Pi_w z + 2\Pi_w^\perp z) + w'). \end{aligned} \tag{46}$$

Now we use $\Pi_w z = (w \cdot \Pi_w z) w$ and find

$$g(z) w \cdot \Pi_w z = (w \cdot \Pi_w z) w \cdot (Q(\Pi_w z + 2\Pi_w^\perp z) + w'). \tag{47}$$

This implies

$$g(z) = w \cdot Q(\Pi_w z + 2\Pi_w^\perp z) + w \cdot w' \tag{48}$$

for all $z$ such that $\Pi_w z \neq 0$. By continuity we conclude that this holds for all $z$, i.e., $g$ is affine. $\quad\square$

Again we need a slight generalization of this lemma.

**Corollary 2.** *Assume that there is a quadratic form $Q$, a vector $0 \neq w \in \mathbb{R}^d$, $w' \in \mathbb{R}^d$, $\alpha, \beta \in \mathbb{R}$ and a continuous function $g : \mathbb{R}^d \to \mathbb{R}$ such that*

$$g(z)(w \cdot z + \beta) = z \cdot Qz + w' \cdot z + \alpha. \tag{49}$$

*Then $g(z) = v \cdot z + c$ for some $v \in \mathbb{R}^d$, $c \in \mathbb{R}$, i.e., $g$ is an affine function.*

*Proof.* Define $\widetilde{g}(\widetilde{z}) = g(\widetilde{z} - \beta w |w|^{-2})$. Then

$$
\begin{aligned}
\widetilde{g}(\widetilde{z})(w \cdot \widetilde{z}) &= g(\widetilde{z} - \beta w |w|^{-2})(w \cdot (\widetilde{z} - \beta w |w|^{-2}) + \beta) \\
&= (\widetilde{z} - \beta w |w|^{-2}) \cdot Q(\widetilde{z} - \beta w |w|^{-2}) + w' \cdot (\widetilde{z} - \beta w |w|^{-2}) + \alpha \\
&= \widetilde{z} \cdot Q\widetilde{z} + \widetilde{w} \cdot \widetilde{z} + \widetilde{\alpha}
\end{aligned} \tag{50}
$$

for some $\widetilde{w}$ and $\widetilde{\alpha}$. So, we conclude from Lemma 5 that $\widetilde{g}$ is affine and thus the same is true for $g$. $\quad\square$

We now discuss a small extension of Lemma 2 by showing that we can complete the squares in (19) which then allows us to obtain a relation that is very similar to (20).

**Lemma 6.** *Assume the general setup as introduced above and also $\Theta^{(i)} \neq \Theta$. Then there is a vector $\mu'^{(i)}$ and a constant $c$ such that*

$$
(z - \mu^{(i)})^\top \Theta^{(i)} (z - \mu^{(i)}) - z^\top \Theta^{(0)} z = (z - \mu'^{(i)})^\top (\Theta^{(i)} - \Theta^{(0)})(z - \mu'^{(i)}) + c. \tag{51}
$$

*Proof.* Using the symmetry of the precision matrices and expanding all the terms, we can express the difference of the left-hand side and the right-hand side of (51) as

$$
\begin{aligned}
&(z - \mu^{(i)})^\top \Theta^{(i)} (z - \mu^{(i)}) - z^\top \Theta^{(0)} z - (z - \mu'^{(i)})^\top (\Theta^{(i)} - \Theta^{(0)})(z - \mu'^{(i)}) - c \\
&= -2z^\top \Theta^{(i)} \mu^{(i)} + 2z^\top (\Theta^{(i)} - \Theta^{(0)}) \mu'^{(i)} + (\mu^{(i)})^\top \Theta^{(i)} \mu^{(i)} - (\mu'^{(i)})^\top (\Theta^{(i)} - \Theta^{(0)}) \mu'^{(i)} - c.
\end{aligned} \tag{52}
$$

Hence, we can set $c$ at the end such the constant terms cancel and all we have to show is that there exists $\mu'^{(i)}$ such that

$$
(\Theta^{(i)} - \Theta)\mu'^{(i)} = \Theta^{(i)} \mu^{(i)}. \tag{53}
$$

This is clearly not true for arbitrary $\mu^{(i)}$ because $\Theta^{(i)} - \Theta$ has only rank 2. However, such a $\mu'^{(i)}$ does exist for $\mu^{(i)} = \eta^{(i)} (B^{(i)})^{-1} e_{t_i}$ (see equation (18)). Indeed, we can simplify using (17) and Lemma 3

$$
(\Theta^{(i)} - \Theta)\mu'^{(i)} = r^{(i)}((r^{(i)})^\top \mu'^{(i)}) - s^{(i)}((s^{(i)})^\top \mu'^{(i)}) \tag{54}
$$

$$
\Theta^{(i)} \mu^{(i)} = \eta^{(i)} (B^{(i)})^\top B^{(i)} (B^{(i)})^{-1} e_{t_i} = \eta^{(i)} (B^{(i)})^\top e_{t_i} = \eta^{(i)} r^{(i)}. \tag{55}
$$

Now there are two cases. When $r^{(i)}$ and $s^{(i)}$ are collinear (but not identical, by assumption) we can choose $\mu'^{(i)}) = \lambda r^{(i)}$ for a suitable $\lambda$. Otherwise, we can pick any vector $\mu'^{(i)}$ such that $\mu'^{(i)} s^{(i)} = 0$, $\mu'^{(i)} r^{(i)} = \eta^{(i)}$. This is always possible for non-collinear vectors (project $r^{(i)}$ on the orthogonal complement of $s^{(i)}$). $\quad\square$

Let us now first discuss a rough sketch of the proof of our main result. This allows us to present the main ideas without discussing all the technical details required for the full proof of the result. We restate the theorem for convenience.

**Theorem 3.** *Assume that $X^{(i)}$ is generated according to a model $((B^{(i)}, \eta^{(i)}, t_i)_{i \in \bar{I}}, f)$ such that the Assumptions 1-4 hold. Then we have identifiability up to linear transformations, i.e., if $((\widetilde{B}^{(i)}, \widetilde{\eta}^{(i)}, \widetilde{t}_i), \widetilde{f})$ generates the same observed distributions $\widetilde{f}(\widetilde{Z}^{(i)}) \stackrel{\mathcal{D}}{=} X^{(i)}$, then their latent dimensions $d$ agree and there is an invertible linear map $T$ such that*

$$
\widetilde{f} = f \circ T^{-1}, \quad \widetilde{Z}^{(i)} = TZ^{(i)}. \tag{5}
$$

*Proof Sketch.* For simplicity of the sketch we ignore shift interventions here. We assume that we have two sets of parameters $((B^{(i)}, t_i)_{i \in \bar{I}}, f)$, $(\widetilde{B}^{(i)}, \widetilde{t}_i)_{i \in \bar{I}}), \widetilde{f})$ generating the same observations. We can relabel the interventions $i$ and the variables $\widetilde{Z}_k$ such that $\widetilde{t}_i = i$ and such that $1, 2, \ldots$ is a valid causal order of the graph. The general idea is to show by induction that $(h_1(z), \ldots, h_k(z))$ is a linear function of $z$. Let us sketch how this is achieved. The key ingredients here are Lemma 2 and Lemma 3. Applying (20) from Lemma 2 in combination with Lemma 3 we obtain

$$
\frac{1}{2} z^\top (\Theta^{(k+1)} - \Theta^{(0)}) z = \frac{1}{2} h(z)^\top (\widetilde{r}^{(k+1)} \otimes \widetilde{r}^{(k+1)} - \widetilde{s}^{(k+1)} \otimes \widetilde{s}^{(k+1)}) h(z) + c^{(k+1)}. \tag{56}
$$

By our assumption that the variables $\widetilde{Z}^{(i)}$ follow the causal order of the underlying graph, only the entries $\widetilde{r}_r^{(k+1)}$ for $r \le k+1$ are non-zero and similarly for $\widetilde{s}^{(k+1)}$. Therefore, the right-hand side is a quadratic form of $(h_1(z), \ldots, h_{k+1}(z))$. Now we apply our induction hypothesis which states that $(h_1(z), \ldots, h_k(z))$ is a linear function of $z$. Then we find expanding the quadratic form on the right-hand side that there is a matrix $Q$, a vector $v$, a constant $\delta$ (which we here assume to be non-zero) and another quadratic form $Q'$ such that

$$
\begin{aligned}
\frac{1}{2}z^\top Q' z &= \frac{1}{2}z^\top Q z + (v \cdot z)h_{k+1}(z) + \delta h_{k+1}^2(z) + c^{(k+1)} \\
&= \frac{1}{2}z^\top Q z + \delta \left( h_{k+1}(z) + \frac{(v \cdot z)}{2\delta} \right)^2 - \frac{(v \cdot z)^2}{4\delta} + c^{(k+1)}.
\end{aligned}
\tag{57}
$$

Now we can apply Corollary 1 to conclude that $h_{k+1}(z)$ is an affine function of $z$. Adding the shifts and considering also the case $\delta = 0$ adds several technical difficulties, which we handle in the full proof. $\qquad\square$

After all those preparations and presenting the proof sketch, we will finally prove our main theorem in full generality.

*Full proof of Theorem 3.* As the proof has to consider different cases and is a bit technical we structure it in a series of steps.

**Labeling assumptions**  We can assume without loss of generality (by relabeling variables and interventions) that the variables $\widetilde{Z}_i$ are ordered such that their natural order $i = 1, 2, \ldots$ is a valid topological order of the underlying DAG and that intervention $1 \le i \le n$ targets node $Z_i$, i.e., $\widetilde{t}_i = i$. We make no assumption on the ordering of the $Z_i$ or targets of $t_i$ which are not necessarily assumed to be pairwise different. Recall that we defined (and showed existence of) $h = \widetilde{f}^{-1} f$ in Lemma 2 (including the fact that their latent dimensions agree).

**Induction claim**  We now prove by induction that $h_k(z)$ is linear for $k \ge 0$. The base case $k = 0$ is trivially true. For the induction step, assume that this is the case for $j \le k$, i.e., $h_j(z) = m_j \cdot z$ for some $m_j \in \mathbb{R}^d$. We define $M_k = (m_1, \ldots, m_k)^\top$ so that we can write concisely

$$
(h_1(z), \ldots, h_k(z))^\top = M_k z.
\tag{58}
$$

Note that since $h$ is surjective $M_k$ has full rank. To prove the induction step, we need to show that there exists $m_{k+1} \in \mathbb{R}^d$ such that $h_{k+1}(z) = m_{k+1} \cdot z$.

**Precision matrix decomposition**  For the DAG $\widetilde{G}$ underlying $\widetilde{Z}_i$, let $\mathrm{PA}_i$ denote the indices of the set of parents of $Z_i$ and let $\overline{\mathrm{PA}}_i = \{i\} \cup \mathrm{PA}_i$. Recall the notation $\widetilde{s}^{(i)} = \widetilde{B}^\top e_i$ (using $\widetilde{t}_i = i$) and $\widetilde{r}^{(i)} = (\widetilde{B}^{(i)})^\top e_i$. Now $\widetilde{s}_j^{(i)} \ne 0$ only holds if $j \in \overline{\mathrm{PA}}_i$ and we ordered the variables such that $\overline{\mathrm{PA}}_i \subset [i]$. In particular, $\widetilde{s}_j^{(i)} = 0$ and $\widetilde{r}_j^{(i)} = 0$ for $j > i$ (interventions do not create new parents). By Lemma 3 we have

$$
(\widetilde{\Theta}^{(i)} - \widetilde{\Theta}^{(0)}) = \widetilde{r}^{(i)} \otimes \widetilde{r}^{(i)} - \widetilde{s}^{(i)} \otimes \widetilde{s}^{(i)}
\tag{59}
$$

We find that

$$
(\widetilde{\Theta}^{(k+1)} - \widetilde{\Theta}^{(0)})_{rs} = \widetilde{r}_r^{(k+1)} \widetilde{r}_s^{(k+1)} - \widetilde{s}_r^{(k+1)} \widetilde{s}_s^{(k+1)} = 0
\tag{60}
$$

if $r > k+1$ or $s > k+1$. In particular, there exists a symmetric $A \in \mathbb{R}^{(k+1) \times (k+1)}$ such that

$$
h(z)^\top (\widetilde{\Theta}^{(i)} - \widetilde{\Theta}^{(0)}) h(z) = \sum_{r,s=1}^{k+1} h_r(z) A_{rs} h_s(z).
\tag{61}
$$

Now we decompose $A$ as

$$
A = \begin{pmatrix} A' & b \\ b^\top & \delta \end{pmatrix}
\tag{62}
$$

for some $A' \in \mathbb{R}^{k \times k}$, $b \in \mathbb{R}^k$ and $\delta \in \mathbb{R}$. Using the induction hypothesis we find

$$
\begin{aligned}
h(z)^\top (\tilde{\Theta}^{(k+1)} - \tilde{\Theta}^{(0)}) h(z) &= \sum_{r,s=1}^{k+1} h_r(z) A_{rs} h_s(z) \\
&= M_k z \cdot A' M_k z + 2 h_{k+1}(z) b^\top M_k z + \delta h_{k+1}(z)^2.
\end{aligned} \tag{63}
$$

We use the notation $v_{:r} \in \mathbb{R}^r$ for the vector $(v_1, \ldots, v_r)^\top$. Then we can write

$$
\eta^{(k+1)} \widetilde{r}^{(k+1)} \cdot h(z) = \eta^{(k+1)} \widetilde{r}_{:k}^{(k+1)} \cdot M_k z + \eta^{(k+1)} \widetilde{r}_{k+1}^{(k+1)} h_{k+1}(z) \tag{64}
$$

Now we have to consider different cases.

**Case $\delta \neq 0$**   We assume first that $\delta \neq 0$. In this case we find using the last two displays

$$
h(z)^\top (\tilde{\Theta}^{(k+1)} - \tilde{\Theta}^{(0)}) h(z) - 2\eta^{(k+1)} \widetilde{r}^{(k+1)} h(z) + c^{(k+1)}
$$

$$
= z \cdot M_k^\top A' M_k z + \delta \left( h_{k+1}(z) + \frac{b^\top M_k z - \eta^{(k+1)} \widetilde{r}_{k+1}^{(k+1)}}{\delta} \right)^2 - \frac{1}{\delta} z \cdot M_k^\top b b^\top M_k z \tag{65}
$$

$$
+ \frac{2}{\delta} \eta^{(k+1)} \widetilde{r}_{k+1}^{(k+1)} b^\top M_k z - \frac{1}{\delta} (\eta^{(k+1)} \widetilde{r}_{k+1}^{(k+1)})^2 - 2\eta^{(k+1)} \widetilde{r}_{:k}^{(k+1)} \cdot M_k z + c^{(k+1)}.
$$

Next we set

$$
g(z) = h_{k+1}(z) + \delta^{-1} b \cdot M_k z - \delta^{-1} \eta^{(k+1)} \widetilde{r}_{k+1}^{(k+1)}. \tag{66}
$$

Looking carefully at (65) we find that there is a symmetric matrix $\tilde{Q}$, a vector $\widetilde{w}$, and a constant $\widetilde{c}$ such that

$$
h(z)^\top (\tilde{\Theta}^{(k+1)} - \tilde{\Theta}^{(0)}) h(z) - 2\eta^{(k+1)} \widetilde{r}^{(k+1)} h(z) = \delta g(z)^2 + z \cdot \tilde{Q} z + \widetilde{w} z + \widetilde{c}. \tag{67}
$$

We claim that $g$ is continuous and surjective. Continuity follows directly from continuity of $h$. To show that $g$ is surjective we can ignore the constant shift, and we note that $\delta^{-1} v^\top M_k z = \delta^{-1} \sum_{r=1}^k v_r h_r(z)$ and thus surjectivity of $h$ implies that $g$ is surjective (for every $c$ pick $z$ such that $h_{k+1}(z) = c$ and $h_r(z) = 0$ for $r \leq k$).

Using (19) from Lemma 2 we conclude that

$$
\delta g(z)^2 = z^\top (\Theta^{(k+1)} - \Theta^{(0)}) z - 2\eta^{(k+1)} r^{(k+1)} z - z \cdot \tilde{Q} z - \widetilde{w} z - \widetilde{c} \tag{68}
$$

$$
= z \cdot Q z + w \cdot z + \alpha \tag{69}
$$

for all $z$ and some symmetric $Q \in \mathbb{R}^{d \times d}$, $w \in \mathbb{R}^d$. Corollary 1 then implies that

$$
h_{k+1}(z) + \delta^{-1} v M_k z + \delta^{-1} \eta^{(k+1)} \widetilde{r}_{k+1}^{(k+1)} = g(z) = v \cdot z + c \tag{70}
$$

for some $v \in \mathbb{R}^d$, $c \in \mathbb{R}$. Thus $h_{k+1}$ is an affine function, i.e., $h_{k+1}(z) = m_{k+1} z + \alpha$. But

$$
0 = \mathbb{E} \widetilde{Z}_{k+1} = \mathbb{E} Z \cdot m_{k+1} + \alpha = \alpha \tag{71}
$$

and therefore $h_{k+1}$ is linear.

**Case $\delta = 0$, $b \neq 0$**   Now we consider the case that $0 = \delta$. Using (63), (64), and (19) we find similarly to above

$$
M_k z \cdot A M_k z + 2 h_{k+1}(z) b^\top M_k z - 2\eta^{(k+1)} \widetilde{r}_{:k}^{(k+1)} \cdot M_k z - 2\eta^{(k+1)} \widetilde{r}_{k+1}^{(k+1)} h_{k+1}(z) + \widetilde{c}^{(k+1)}
$$

$$
= z^\top (\Theta^{(k+1)} - \Theta^{(0)}) z - 2\eta^{(k+1)} r^{(k+1)} z \tag{72}
$$

Collecting all terms we find that $h_{k+1}$ satisfies a relation of the type

$$
h_{k+1}(z)(w \cdot z + \beta) = z \cdot Q z + w' \cdot z + \alpha \tag{73}
$$

for suitable $Q$, $\beta$, $w$, $w'$, and $\alpha$. Note in particular that $w^\top = 2b^\top M_k \neq 0$ because $M_k$ has full rank. If $w \neq 0$ then Corollary 2 implies that $h_k(z)$ is affine and, as argued above, we conclude that $h_k$ is linear.

**Case $\delta = 0$, $b = 0$**  It remains to address the case $b = 0$ and $\delta = 0$. Going back to the definition of $b$ and $\delta$ (see (61) and (62)) we find

$$(b^\top, \delta)^\top = \widetilde{r}^{(k+1)}_{:(k+1)} \widetilde{r}^{(k+1)}_{k+1} - \widetilde{s}^{(k+1)}_{:(k+1)} \widetilde{s}^{(k+1)}_{k+1} \tag{74}$$

Now $\delta = 0$ implies $\widetilde{s}^{(k+1)}_{k+1} = \widetilde{r}^{(k+1)}_{k+1}$ (because they are both positive). But then we conclude $\widetilde{r}^{(k+1)} = \widetilde{s}^{(k+1)}$ from $b = 0$ and $\widetilde{r}^{(k+1)}_r = \widetilde{s}^{(k+1)}_r = 0$ for $r > k + 1$. This implies

$$\widetilde{\Theta}^{(k+1)} - \widetilde{\Theta}^{(0)} = \widetilde{r}^{(k+1)} \otimes \widetilde{r}^{(k+1)} - \widetilde{s}^{(k+1)} \otimes \widetilde{s}^{(k+1)} = 0 \tag{75}$$

That is, this implies that intervention $k + 1$ is a pure shift intervention, i.e., $B = B^{(k+1)}$ and $\eta^{(k+1)} \neq 0$ (otherwise we do not actually intervene). In this case, we conclude from Lemma 2 and Lemma 6 (if $\Theta^{(k+1)} \neq \Theta$)

$$\eta^{(k+1)} \widetilde{r}^{(k+1)} h(z) + c^{(k+1)} = (z - \mu'^{(k+1)})^\top (\Theta^{(k+1)} - \Theta)(z - \mu'^{(k+1)}) + c. \tag{76}$$

But then $\eta^{(k+1)} \widetilde{r}^{(k+1)} \nabla h(\mu'^{(k+1)}) = 0$, this implies that $\nabla h$ is not invertible which is a contradiction to $h$ being a diffeomorphism. Thus, we conclude that also $\Theta^{(k+1)} = \Theta$, i.e., intervention $k + 1$ is also a pure shift intervention for the $Z$ variables. But then we find

$$\widetilde{r}^{(k+1)}_{k+1} h_{k+1}(z) + \widetilde{r}^{(k+1)}_{:k} M_k z = (\widetilde{\eta}^{(k+1)})^{-1}(\eta^{(k+1)} r^{(k+1)} z + c^{(k+1)}). \tag{77}$$

So we finally conclude that also in this case $h_{k+1}$ is a linear function, this ends the proof. $\qquad\square$

## B   From Linear Identifiability to Full Identifiability

In this section, we provide the proof of Theorem 1 which is a combination of our linear identifiability result Theorem 3 proved in the previous section and the main result of [84] which shows identifiability for linear mixing $f$. However, we need to carefully address their normalization choices and in addition consider shifts.

We emphasize that the full identifiability results could also be derived from our proof of Theorem 3 by carefully considering ranks of quadratic forms and showing that we can inductively identify source nodes. We do not add these arguments as our reasoning is already quite complex and because for linear mixing functions this is not substantially different from the original proof in [84].

For the convenience of the reader, we will now restate the main results of [84]. For our work it is convenient to slightly deviate from their conventions, i.e., we define the causal order of a graph by $i \prec j$ iff $i \in \mathrm{an}(j)$, in particular $i \to j$ implies $i \prec j$.

We rephrase their results using our notation. Let $S(G)$ denote the set of permutations of the vertices of $G$ that preserve the causal ordering, i.e., all permutations $\rho$ such that $\rho(i) < \rho(j)$ if $i \prec j$. They consider linear mixing functions as specified next.

**Assumption 5.** *Assume $f(z) = Lz$ for some matrix $L \in \mathbb{R}^{d' \times d}$ with trivial kernel and pseudoinverse $H = L^+$ such that the largest absolute value of each row of $H$ is one and the leftmost is positive.*

Then the following result holds.

**Theorem 5** (Linear setting, Theorems 1 and 2 in [84]). *Suppose latent variables $Z^{(i)}$ are generated following Assumptions 2 and 3 where $\eta^{(i)} = 0$ (i.e., no shift) with DAG $G$ and all $A^{(i)}$ are lower triangular. Suppose in addition that Assumption 4 holds and the mixing function satisfies Assumption 5. Then we can identify intervention targets and the partial order $\prec_G$ of the underlying DAG $G$ up to permutation of the node labels. If we in addition assume that all interventions are perfect the problem is identifiable in the following sense. For every representation $(\widetilde{B}^{(i)}, \widetilde{H})$ such that all $\widetilde{B}^{(i)}$ are lower triangular and $\widetilde{L}(\widetilde{Z}^{(i)}) \overset{\mathcal{D}}{=} X^{(i)}$ where $\widetilde{L} = \widetilde{H}^+$ there is a permutation $\sigma \in S(G)$ such that*

$$\widetilde{H} = P_\sigma^\top H, \quad \widetilde{B}^{(i)} = P_\sigma^\top B^{(i)} P_\sigma. \tag{78}$$

**Remark 5.** *A few remarks are in order.*

   1. *Compared to their statement in Theorem 2 we replaced $P_\sigma$ by $P_\sigma^\top$ because we used the transposed convention for the permutation matrices.*

2. *Note that their result per se doesn't mention identifiability up to scaling but that's because part of their assumptions involve normalizing $f$. Since we don't normalize the linear map $f$, we add scaling in the theorem statement.*

3. *For the case of imperfect interventions, their result talks about identifiability of the transitive closure $\overline{G}$ of the graph $G$, but this is the same as identifiability of the partial order $\prec_G$ that we state here.*

4. *We assume that interventions are not only acting on the noise but do change the corresponding row of $A$. Note that this is equivalent to their genericity assumption which states that*

$$r^{(i)} = (B^{(i)})^\top e_{t_i} = (\mathrm{Id} - A^{(i)})_{t_i \cdot} (D^{(i)})_{t_i t_i}^{-\frac{1}{2}} \tag{79}$$

$$s^{(i)} = (B^{(0)})^\top e_{t_i} = (\mathrm{Id} - A^{(0)})_{t_i \cdot} (D^{(0)})_{t_i t_i}^{-\frac{1}{2}} \tag{80}$$

*are linearly independent. Indeed, this holds iff $A_{t_i \cdot}^{(0)} \neq A_{t_i \cdot}^{(i)}$ (by acyclicity $A_{t_i t_i}^{(i)} = 0$).*

Let us provide a brief intuition why this result holds. The key ingredient is the same as in our proof of linear identifiability above, namely, Lemma 3 which reads

$$\Theta^{(i)} - \Theta^{(0)} = r^{(i)} \otimes r^{(i)} - s^{(i)} \otimes s^{(i)}. \tag{81}$$

This expression has rank 2 if $t_i$ is not a source node in the causal graph and the intervention is not a pure noise intervention because $s^{(i)}$ and $r^{(i)}$ are linearly independent as explained in the remark above. On the other hand, this expression has rank one for source nodes because $s^{(i)} \propto e_{t_i}$ in this case and the same is true for $r^{(i)}$. This allows us to identify interventions on source nodes. Moreover, $\Theta^{(i)} - \Theta^{(k)}$ has rank 2 if intervention $i$ and $k$ target different source nodes and rank 1 if they target the same source node. Thus we can also identify the set of interventions targeting the same source node. One can complete the proof by an induction argument based on removing source nodes from the graph and studying the effect on the precision matrix of the latent variables.

We are now ready to prove our main theorems.

**Theorem 1.** *Suppose we are given distributions $X^{(i)}$ generated using a model $((B^{(i)}, \eta^{(i)}, t_i)_{i \in \bar{I}}, f)$ such that Assumptions 1-4 hold and such that all interventions $i$ are perfect. Then the model is identifiable up to permutation and scaling, i.e., for any model $((\widetilde{B}^{(i)}, \widetilde{\eta}^{(i)}, \widetilde{t}_i)_{i \in \bar{I}}, \widetilde{f})$ that generates the same data, the latent dimension $d$ agrees and there is a permutation $\omega \in S_d$ (and associated permutation matrix $P_\omega$) and an invertible pointwise scaling matrix $\Lambda \in \mathrm{Diag}(d)$ such that*

$$\widetilde{t}_i = \omega(t_i), \quad \widetilde{B}^{(i)} = P_\omega^\top B^{(i)} \Lambda^{-1} P_\omega, \quad \widetilde{f} = f \circ \Lambda^{-1} P_\omega, \quad \widetilde{\eta}^{(i)} = \eta^{(i)}. \tag{3}$$

*This in particular implies that*

$$\widetilde{Z}^{(i)} \stackrel{\mathcal{D}}{=} P_\omega^\top \Lambda Z^{(i)} \tag{4}$$

*and we can identify the causal graph $G$ up to permutation of the labels.*

*Proof of Theorem 1.* Suppose there are two representations $((B^{(i)}, \eta^{(i)}, t_i)_{i \in I}, f)$ and $((\widetilde{B}^{(i)}, \widetilde{\eta}^{(i)}, \widetilde{t}_i)_{i \in I}, \widetilde{f})$ of the distributions $X^{(i)}, i \geq 0$. As explained in the proof of Theorem 3 we can assume by relabeling the nodes that the matrices $\widetilde{B}^{(i)}$ are lower triangular and the corresponding DAG $\widetilde{G}$ has the property that for every edge $i \rightarrow j$ we have $i < j$. We now apply Theorem 3 and find that there is an invertible linear map $T$ such that $\widetilde{f} = f \circ T$ and $T^{-1} Z^{(i)} \stackrel{\mathcal{D}}{=} \widetilde{Z}^{(i)}$. One minor difference to the setting in [84] is that we in addition consider shift interventions. They are simpler to analyze than a change of variance, but we can avoid additional arguments with the following trick. If $\Sigma^{(i)} = \Sigma^{(0)}$ for some $i$ we replace $Z^{(i)} = N(\mu^{(i)}, \Sigma^{(i)})$ by $Z'^{(i)} = N(0, \Sigma^{(i)} + \mu^{(i)}(\mu^{(i)})^\top)$, otherwise we set $Z'^{(i)} = N(0, \Sigma^{(i)})$ and similarly for $\widetilde{Z}$. Then $T^{-1}(Z^{(i)}) \stackrel{\mathcal{D}}{=} \widetilde{Z}^{(i)}$ implies $T^{-1}(Z'^{(i)}) \stackrel{\mathcal{D}}{=} \widetilde{Z}'^{(i)}$. Using (13) and (17) it can be shown that this distribution corresponds to a change of variance of the noise, i.e., a change of $D_{t_i t_i}^{(i)}$. This can also be seen directly from the relation $Z'^{(i)} \stackrel{\mathcal{D}}{=} (B^{(i)})^{-1}(\epsilon + \eta^{(i)} \epsilon' e_{t_{(i)}})$, where $\epsilon' \sim N(0, 1)$ and

independent of $\epsilon$. We can now apply Theorem 5 to the primed distributions. We first find a unique invertible $\Lambda \in \mathrm{Diag}(d)$ such that $\Lambda T$ has largest absolute value entry 1 in every row (and in case of ties the leftmost entry is 1). We can also find a permutation $\rho \in S_d$ such that $\bar{B}^{(i)} = P_\rho B^{(i)} \Lambda^{-1} P_\rho^\top$ is lower triangular for all $i$ (note the distinction between $\bar{B}^{(i)}$ and $\widetilde{B}^{(i)}$). Indeed, this is possible since the underlying DAG $G$ is acyclic by assumption, interventions do not add edges, and $\Lambda$ is diagonal. Then we find

$$(T^{-1}\Lambda^{-1}P_\rho^\top)(\bar{B}^{(i)})^{-1}\epsilon \stackrel{\mathcal{D}}{=} (T^{-1}\Lambda^{-1}P_\rho^\top)(\bar{B}^{(i)})^{-1}P_\rho\epsilon \tag{82}$$

$$= T^{-1}(B^{(i)})^{-1}\epsilon \tag{83}$$

$$\stackrel{\mathcal{D}}{=} (\widetilde{B}^{(i)})^{-1}\epsilon. \tag{84}$$

Here the last step follows from the fact that $T^{-1}Z^{(i)} \stackrel{\mathcal{D}}{=} \widetilde{Z}^{(i)}$ which implies that the centered distributions agree.

Now we can apply Theorem 5 (with $B$ replaced by $\widetilde{B}$ and identity mixing) and find that for the alternative representation in terms of $\bar{B}$

$$P_\rho \Lambda T = P_\sigma^\top \mathrm{Id}, \quad \bar{B}^{(i)} = P_\sigma^\top \widetilde{B}^{(i)} P_\sigma \tag{85}$$

for some $\rho \in S(\widetilde{G})$ because by assumption $\bar{B}^{(i)}$ and $\widetilde{B}$ are lower triangular. This implies that

$$T = \Lambda^{-1}P_\rho^\top P_\sigma^\top, \tag{86}$$

$$B^{(i)} = P_\rho^\top \bar{B}^{(i)} P_\rho \Lambda = P_\rho^\top P_\sigma^\top \widetilde{B}^{(i)} P_\sigma P_\rho \Lambda. \tag{87}$$

To summarize, there is a permutation $\omega = (\rho \circ \sigma)^{-1}$ (note that $P_\sigma P_\rho = P_{\rho \circ \sigma}$) and a diagonal matrix $\Lambda$ such that

$$B^{(i)} = P_\omega \widetilde{B}^{(i)} P_\omega^\top \Lambda, \quad T = \Lambda^{-1}P_\omega, \quad Z^{(i)} = T\widetilde{Z}^{(i)} = \Lambda^{-1}P_\omega \widetilde{Z}^{(i)}, \tag{88}$$

We also find the relation $\omega(t_i) = \widetilde{t}_i$ from here because $P_\omega e_i = e_{\omega^{-1}(i)}$. Equating the means of $T^{-1}Z^{(i)}$ and $\widetilde{Z}^{(i)}$ we find

$$\widetilde{\eta}^{(i)}(\widetilde{B}^{(i)})^{-1}e_{\widetilde{t}_i} = \eta^{(i)}T^{-1}(B^{(i)})^{-1}e_{t_i}. \tag{89}$$

Multiplying by $\widetilde{B}^{(i)}$ we find

$$\widetilde{\eta}^{(i)}e_{\widetilde{t}_i} = \eta^{(i)}\widetilde{B}^{(i)}T^{-1}(B^{(i)})^{-1}e_{t_i} = \eta^{(i)}P_\omega^\top e_{t_i} \tag{90}$$

and we conclude that $\widetilde{\eta}^{(i)} = \eta^{(i)}$.

$\square$

**Theorem 2.** *Suppose we are given the distributions $X^{(i)}$ generated using a model $((B^{(i)}, \eta^{(i)}, t_i)_{i \in \bar{I}}, f)$ with causal graph $G$ such that the Assumptions 1-4 hold and none of the interventions is a pure noise intervention. Then for any other model $((\widetilde{B}^{(i)}, \widetilde{\eta}^{(i)}, \widetilde{t}_i)_{i \in \bar{I}}, \widetilde{f})$ with causal graph $\widetilde{G}$ generating the same observations the latent dimension $d$ agrees and there is a permutation $\omega \in S_d$ such that $\widetilde{t}_i = \omega(t_i)$ and $i \prec_{\widetilde{G}} j$ iff $\omega(i) \prec_G \omega(j)$, i.e., $\prec_G$ can be identified up to a permutation of the labels.*

*Proof of Theorem 2.* Suppose there are two representations $((B^{(i)}, \eta^{(i)}, t_i)_{i \in I}, f)$ and $((\widetilde{B}^{(i)}, \widetilde{\eta}^{(i)}, \widetilde{t}_i)_{i \in I}, \widetilde{f})$ of the observational distribution. As shown in Theorem 3 there is a linear map such that $T^{-1}Z^{(i)} = \widetilde{Z}^{(i)}$. Then the same is true for the transformed variables as explained above. Viewing $\widetilde{Z}^{(i)}$ as new observations obtained through a linear $T$ mixing we conclude from Theorem 5 that $\prec_G$ and the intervention targets are identifiable up to relabeling. $\square$

# C  A comparison to related works

In this section, we will discuss the relation of our results to the most relevant prior works and put them into context.

**[84]**  This work considers linear mixing functions $f$, and we directly generalize their work to non-linear $f$. In particular, their techniques are linear-algebraic and do not generalize to non-linear $f$ (see more technical details in proof intuition in Section 4); we handle this using a mix of statistical and geometric techniques. Notably, their finding that the linearly mixed latent variables are recoverable after observing one intervention per latent node is closely related to an earlier result in domain generalization, which showed that a number of environments equal to the number of "non-causal" latent dimensions can ensure recovery of the remaining latents [71], and later work showing that this requirement can be reduced further under linear mixing with additional modeling structure [11, 72]. This connection is not coincidental—that analysis was for invariant prediction with latent variables, which was originally presented as a method of causal inference using distributions resulting from unique interventions [62]. Later this approach was applied for the purpose of robust prediction in nonlinear settings [25], but the identifiability of the latent causal structure under general nonlinear mixing via interventional distributions remained an open question until this work.

**[88]**  This work also considers linear mixing $f$, but allow for non-linearity in the latent variables. They use score functions to learn the model and they raise as an open question the setting of non-linear mixing, which we study in this work. While the models are not directly comparable, our model is more akin to real-world settings since it's not likely that high-level latent variables are linearly related to the observational distribution, e.g. pixels in an image are not linearly related to high-level concepts. Moreover, Gaussian priors and deep neural networks are extensively used in practice, and our theory as well as experimental methodology apply to them.

**[48]**  While they do not discuss interventions, this work considers the setting of multi-environment CRL with Gaussian priors and their results can be applied to interventional learning. When applied to interventional data as in our setting (as also shown in [84, Appendix D]), they require additional restrictive assumptions such as $d = d'$, a bijective mixing $f$ and $e \leq d$ where $e$ is the number of edges in the underlying causal graph whereas we make no such restrictions on $d, f$ and $e$ (therefore, the maximum value of $e$ can be as large as $\approx d^2/2$ in our setting). And when these assumptions are satisfied, they require $2d$ interventions whereas we show that $d$ interventions are sufficient as well as necessary.

**[2]**  This work also considers the setup of interventional causal representation learning. However, their setting is different in various ways and we now outline the key differences.

1. Their main results assume that the mixing $f$ is an *injective polynomial* of finite degree $p$ (see Assumption 2 in their paper). The injectivity requires their coefficient matrix to be full rank, which in turn implies $d' \geq \sum_{r=0}^{p} \binom{r+d-1}{d-1} = \binom{p+d}{d}$ (see the discussion below Assumption 2 in their work). This means that we cannot set the degree of the decoder polynomial to be arbitrarily large for fixed output dimension $d'$ (which is required for universal approximation). In contrast, we only require $d' \geq d$. For a general discussion of the relation between approximability and identifiability we refer to Appendix E.

2. All their results for general nonlinear mixing functions rely on deterministic do-interventions which is a type of hard intervention that assigns a constant value to the target. However, we focus on randomized soft interventions (and also allow shift interventions which have found applications [73, 58]), which as also pointed out by [84, 87] is less restricted. Moreover, they require multiple interventions for each latent variable (as they use an $\epsilon$-net argument) to show approximate identifiability, while the structural assumptions on the causal model allow us to show full identifiability with just one intervention for each latent variable. Thus, their setting is not directly comparable to ours and neither is strictly more general than the other. In particular, their proof techniques and experimental methodology don't translate to our setting, and we use completely different ideas for our proofs and experiments.

## D   Counterexamples and Discussions

In this section, we first present several counterexamples to relaxed versions of our main results in Section D.1 and then provide the missing proofs in Section D.2.

## D.1 Main counterexamples

In this section, we will discuss the optimality and limitations of various assumptions of our main results. In particular, we consider the number of interventions, the distribution of the latent variables, and the types of interventions separately.

**Number of interventions** Our main results all require $d$ interventions, i.e., one intervention per node. This is necessary even in the simpler setting of linear $f$ as was shown in [84, Proposition 4], directly implying it is also necessary for the more general class of non-linear $f$ that we handle here. Therefore, the number of interventions in our main theorems 1, 2 is both necessary and sufficient. Going a step further, it is natural to ask whether Theorem 3 remains true for less than $d$ interventions. However, as we show next, $d-2$ interventions are not sufficient.

**Fact 1.** *Suppose we are given distributions $X^{(i)}$ generated by $((B^{(i)}, \eta^{(i)}, t_i)_{i \in \bar{I}}, f)$ satisfying Assumption 1-3. If the number of interventions satisfies $|I| \leq d-2$ it is not possible to identify $f$ up to linear maps. Consider, e.g., $d = 2$ and $Z = N(0, \mathrm{Id})$ and no interventions. Then $h(Z) \stackrel{\mathcal{D}}{=} Z$ for (nonlinear) radius dependent rotations as defined in [29, 9].*

Let us next consider the case with a total of $d$ environments which corresponds to $d-1$ interventional distributions plus one observational distribution. We can provide a non-identifiability result for a general class of heterogeneous latent variable models that satisfy an algebraic property.

**Lemma 7.** *Assume that $d = 2$ and $Z^{(0)} \sim N(0, \Sigma^{(0)})$, $Z^{(1)} \sim N(0, \Sigma^{(1)})$ and $\Sigma^{(1)} \succ \Sigma^{(0)}$ or $\Sigma^{(1)} \prec \Sigma^{(0)}$ in Löewner order (i.e., the difference is positive definite). Then there is a non-linear map $h$ such that $h(Z^{(i)}) \stackrel{\mathcal{D}}{=} Z^{(i)}$ for $i = 0, 1$.*

We provide a proof of this lemma in Appendix D.2. Note that this non-identifiability lemma requires the assumption on $\Sigma^{(i)}$. In Lemma 3 in Appendix A we have seen that for the interventions considered in this work the relation $\Sigma^{(i)} > \Sigma^{(0)}$ or $\Sigma^{(i)} < \Sigma^{(0)}$ generally does not hold. This is some weak indication that for Theorem 3, $d-1$ interventions might be sufficient. We leave this for future work.

As an additional note, in the special case when $f$ is a permutation matrix, we are in the setting of traditional causal discovery and here, $d-1$ interventions are necessary and sufficient [10, 18, 84].

**Type of intervention** Even when we restrict ourselves to only linear mixing functions as considered as in [84], we need perfect interventions in Theorem 1. Otherwise, only the weaker identifiability guarantees in Theorem 2 can be obtained. This is shown in [84] for linear $f$. Therefore, in the more general setting of nonlinear $f$, we cannot hope to obtain Theorem 1 with imperfect interventions, showing that this assumption is needed.

Next, we clarify that for the identifiability result in Theorem 2 , the condition that the interventions are not pure noise interventions is also necessary.

**Fact 2.** *Recall that we defined $B^{(i)} = (D^{(i)})^{-\frac{1}{2}}(\mathrm{Id} - A^{(i)})$. Suppose $X^{(i)}$ is generated by $((B^{(i)}, \eta^{(i)}, t_i)_{i \in \bar{I}}, f)$ satisfying Assumptions 1-4 where all interventions are pure noise interventions. By definition, this implies $A^{(i)} = A$ for all $i$. Consider any matrix $\widetilde{A}$ such that the corresponding DAG is acyclic. Then $((\widetilde{B}^{(i)}, \eta^{(i)}, t_i)_{i \in \bar{I}}, \widetilde{f})$ with $\widetilde{B}^{(i)} = (D^{(i)})^{-\frac{1}{2}}(\mathrm{Id} - \widetilde{A})$ for all $i$ and*

$$\widetilde{f} = f \circ (B^{-1}\widetilde{B}) = f \circ (\mathrm{Id} - A)^{-1}(D^{(0)})^{-\frac{1}{2}}(D^{(0)})^{\frac{1}{2}}(\mathrm{Id} - \widetilde{A}) = f \circ (\mathrm{Id} - A)^{-1}(\mathrm{Id} - \widetilde{A}) \quad (91)$$

*generates the same distributions $X^{(i)}$. Indeed,*

$$\begin{aligned}
\widetilde{f}(\widetilde{Z}^{(i)}) &= f \circ (B^{-1}\widetilde{B})(\widetilde{B}^{(i)})^{-1}(\epsilon + \eta^{(i)}e_{t_i}) \\
&= f((\mathrm{Id} - A)^{-1}(\mathrm{Id} - \widetilde{A})(\mathrm{Id} - \widetilde{A})^{-1}(D^{(i)})^{\frac{1}{2}}(\epsilon + \eta^{(i)}e_{t_i})) \\
&= f((\mathrm{Id} - A)^{-1}(D^{(i)})^{\frac{1}{2}}(\epsilon + \eta^{(i)}e_{t_i})) \qquad\qquad (92) \\
&= f((B^{(i)})^{-1}(\epsilon + \eta^{(i)}e_{t_i})) \\
&= f(Z^{(i)}).
\end{aligned}$$

*This implies that any underlying causal graph could generate the observations. In other words, for pure noise interventions we can identify the scale of the noise variables, but we obtain no intervention*

*about the causal variables and structure (exactly because the interventions do not target the actual causal variables but the noise variables).*

Finally, we remark that in contrast to the case with linear mixing functions, we need to exclude non-stochastic hard interventions for linear identifiability.

**Lemma 8.** *Consider $d = 2$ and $Z^{(0)} = N(0, \mathrm{Id})$ and the non-stochastic hard interventions $\mathrm{do}(Z_1 = 0)$ and $\mathrm{do}(Z_2 = 0)$ resulting in the degenerate Gaussian distributions $Z^{(1)} = N(0, e_2 \otimes e_2)$ and $Z^{(2)} = N(0, e_1 \otimes e_1)$. Then there is a nonlinear $h$ such that $h(Z^{(i)}) \overset{\mathcal{D}}{=} Z^{(i)}$ for all $i$.*

Intuitively, to show this, we choose $h$ to be identity close to the supports $\{Z_1 = 0\}$ and $\{Z_2 = 0\}$ but some nonlinear measure preserving deformation in the four quadrants. The full argument can be found in Appendix D.2. A similar counterexample was discussed in [2]. However, here we in addition constrain the distribution of the latent variables, which adds a bit of complexity.

**Distribution of the Noise Variables**  Our key additional assumption compared to the setting considered in [84, 2, 88] is that we assume that the latent variables are Gaussian which allows us to put only very mild assumptions on the mixing function $f$. However, we do this in a manner so that we still preserve universal representation guarantees.

[88] consider arbitrary noise distributions but restrict to linear mixing functions. We believe that our result can be generalized to more general distributions of the noise variables, e.g., exponential families, but those generalizations will require different proof strategies, which we leave for future work. However, the result will not be true for arbitrary noise distributions, as the following lemma shows.

**Lemma 9.** *Consider $Z^{(0)} = (\epsilon_1, \epsilon_2)^\top$, $Z^{(1)} = (\epsilon'_1, \epsilon_2)$, and $Z^{(2)} = (\epsilon_1, \epsilon'_2)$ where $\epsilon_1, \epsilon_2 \sim \mathcal{U}([-1, 1])$ and $\epsilon'_1, \epsilon'_2 \sim \mathcal{U}([-3, 3])$. Also define $\widetilde{Z}^{(0)} = (\epsilon_1, \epsilon_1 + \epsilon_2)$, $\widetilde{Z}^{(1)} = (\epsilon'_1, \epsilon'_1 + \epsilon_2)$, and $\widetilde{Z}^{(2)} = (\epsilon_1, \epsilon'_2)$. Then there is $h$ such that*

$$h(Z^{(i)}) \overset{\mathcal{D}}{=} \widetilde{Z}^{(i)}. \tag{93}$$

A proof of this lemma can be found in Appendix D.2. The result shows that even with $d$ perfect interventions and linear SCMs it is not possible to identify the underlying DAG (empty graph for $Z$ and $\widetilde{Z}_1 \to \widetilde{Z}_2$) and we also cannot identify $f$ up to linear maps. While our example relies on distributions with bounded support, we conjecture that full support is not sufficient for identifiability.

**Structural Assumptions**  A final assumption used in our results is the restriction to linear SCMs. This is used for most works on causal representation learning (see Section 2), even when they restrict to linear mixing functions. Although this may potentially be a nontrivial restriction, the advantage is that the simplicity of the latent space will enable us to meaningfully probe the latent variables, intervene and reason about them. Nevertheless, it's an interesting direction to study to what generality this can be relaxed. Indeed, even for the arguably simpler problem of causal structure learning (i.e., when $Z^{(i)}$ is observed) there are many open questions when moving beyond additive noise models. We leave this for future work.

### D.2 Technical constructions

In this section, we provide the proofs for the counterexamples in Section D.1. For the first two counterexamples we construct functions $h$ such that $h(Z^{(i)}) \overset{\mathcal{D}}{=} Z^{(i)}$ by setting $h = \Phi_t$ where $\Phi_t$ is defined as the flow of a suitable vectorfield $X$, i.e.,

$$\partial_t \Phi_t(z) = X(\Phi_t(z)). \tag{94}$$

This is a common technique in differential geometry and was introduced to construct counterexamples to identifiability ICA with volume preserving functions in [9, 8]. To find a suitable vectorfield $X$ we rely on the fact that the density $p_t$ of $\mathbb{P}_t = (\Phi_t)_* \mathbb{P}$ satisfies the continuity equation

$$\partial_t p_t(z) + \mathrm{Div}(p_t(z) X(z)) = 0. \tag{95}$$

In particular $\mathbb{P}_t = \mathbb{P}$ holds iff

$$\mathrm{Div}(p_t(z) X(z)) = 0. \tag{96}$$

Therefore it is sufficient to find a vectorfield $X$ such that $\mathrm{Div}(p^{(i)}X) = 0$ for all environments $i$ to construct a suitable $h = \Phi_1$.

**Lemma 7.** *Assume that $d = 2$ and $Z^{(0)} \sim N(0, \Sigma^{(0)})$, $Z^{(1)} \sim N(0, \Sigma^{(1)})$ and $\Sigma^{(1)} \succ \Sigma^{(0)}$ or $\Sigma^{(1)} \prec \Sigma^{(0)}$ in Löewner order (i.e., the difference is positive definite). Then there is a non-linear map $h$ such that $h(Z^{(i)}) \overset{\mathcal{D}}{=} Z^{(i)}$ for $i = 0, 1$.*

*Proof of Lemma 7.* Let us first discuss the high-level idea of the proof. The general strategy is to construct a vectorfield $X$ whose flow preserves $Z^{(i)}$ for $i = 0, 1$. This holds if and only if the densities $p_0$, $p_1$ of $Z^{(0)}$, $Z^{(1)}$ satisfy

$$\mathrm{Div}(p_0 X) = \mathrm{Div}(p_1 X) = 0 \tag{97}$$

Using $\mathrm{Div}(fX) = \nabla f \cdot X + f \, \mathrm{Div} \, X$ we find

$$X \cdot \nabla(\ln p_0(z) - \ln p_1(z)) = \frac{X \cdot \nabla p_0(z)}{p_0(z)} - \frac{X \cdot \nabla p_1(z)}{p_1(z)} - \frac{p_0(z) \, \mathrm{Div} \, X}{p_0(z)} + \frac{p_1(z) \, \mathrm{Div} \, X}{p_1(z)} = 0. \tag{98}$$

We conclude that any $X$ satisfying (97) must be orthogonal to $\nabla(\ln p_0(z) - \ln p_1(z))$ and thus parallel to the level lines of $\ln p_0(z) - \ln p_1(z)$. This already fixes the direction of $X$ and the magnitude can be inferred from (97). To simplify the calculations we can first linearly transform the data so that the directions of $X$, i.e., equivalently the level lines of $\ln p_0(z) - \ln p_1(z)$ have a simple form. We assume that $\Sigma_0 \prec \Sigma_1$. Clearly, it is sufficient to show the result for any linear transformation of the Gaussian distributions $Z^{(i)}$. Note that generally for $G \sim N(\mu, \Sigma)$ we have $AG \sim N(A\mu, A\Sigma A^\top)$ Applying $\Sigma_0^{-\frac{1}{2}}$ we can reduce the problem to $\mathrm{Id} \prec \Sigma_1' = \Sigma_0^{-\frac{1}{2}} \Sigma_1 \Sigma_0^{-\frac{1}{2}}$. Diagonalizing $\Sigma_1'$ by $U \in \mathrm{SO}(d)$ it is sufficient to consider $\mathrm{Id} \prec \Lambda = U \Sigma_1' U^\top$ where $\Lambda = \mathrm{Diag}(\lambda_1, \lambda_2)$. Finally, we can rescale $z_1$ and $z_2$ such that the resulting covariance matrices $\Lambda^0 = \mathrm{Diag}(\lambda_1^0, \lambda_2^0)$, $\Lambda^1 = \mathrm{Diag}(\lambda_1^1, \lambda_2^1)$ satisfy

$$\frac{1}{\lambda_1^1} - \frac{1}{\lambda_1^0} = \frac{1}{\lambda_2^1} - \frac{1}{\lambda_2^0} = 1. \tag{99}$$

Thus, it is sufficient to show the claim for such covariances $\Lambda^0$, $\Lambda^1$ and the result for arbitrary covariances follows by applying suitable linear transformation.

For such covariances we find for some constant $c$

$$\ln(p_0(z)) - \ln(p_1(z)) = -\frac{z_1^2}{2\lambda_1^0} - \frac{z_2^2}{2\lambda_2^0} + \frac{z_1^2}{2\lambda_1^1} + \frac{z_2^2}{2\lambda_2^1} + c = z_1^2 + z_2^2 + c. \tag{100}$$

The level lines are circles. Thus, we consider the vector field

$$X_0(z) = \begin{pmatrix} -z_2 \\ z_1 \end{pmatrix}. \tag{101}$$

We see directly that $\mathrm{Div} \, X_0 = 0$. We consider the Ansatz $X(z) = f(|z|) X_0(z) / p_0(z)$. We now fix the radial function $f : \mathbb{R}_+ \to \mathbb{R}$ such that it has compact support away from 0. We find using $\mathrm{Div} \, X_0 = 0$

$$\mathrm{Div}(X(z) p_0(z)) = \mathrm{Div}(X(z) p_0(z)) \tag{102}$$

$$= X_0(z) \cdot (\nabla |z|) f'(z) \tag{103}$$

$$= \begin{pmatrix} -z_2 \\ z_1 \end{pmatrix} \cdot \frac{z}{|z|^2} f'(z) \tag{104}$$

$$= 0. \tag{105}$$

Note, that close to 0 where $|z|$ is not differentiable the vector field vanishes by our construction of $f$. We find similarly using (100)

$$\mathrm{Div}(X(z) p_1(z)) = X_0(z) \cdot \nabla \left( f(|z|) \frac{p_1(z)}{p_0(z)} \right) \tag{106}$$

$$= X_0(z) \cdot \nabla \left( f(|z|) e^{-|z|^2 - c} \right) \tag{107}$$

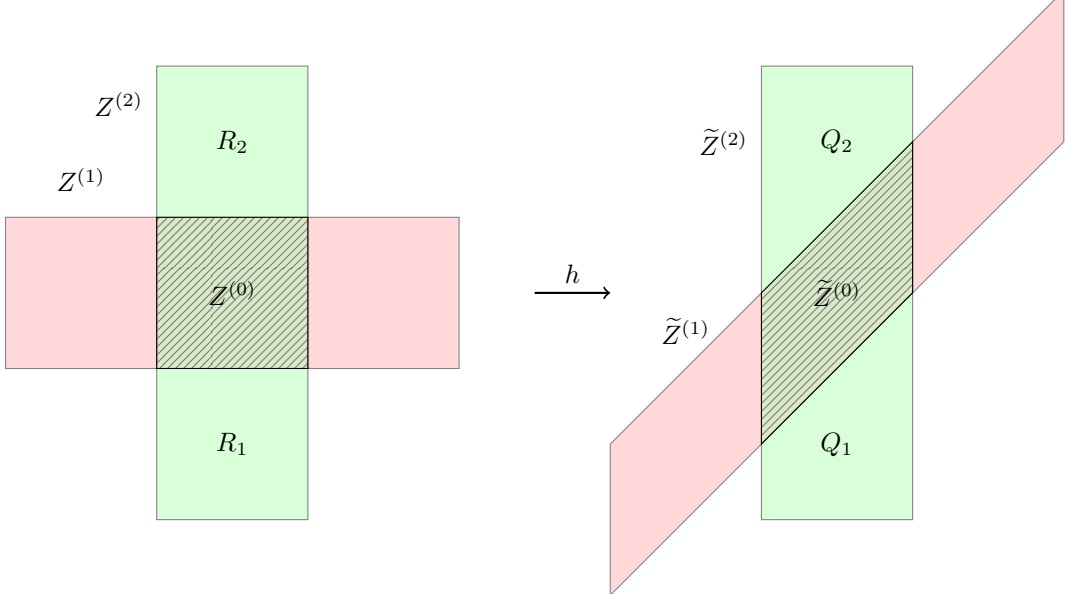

Figure 4: Sketch of the setting in Lemma 9. The map $h$ agrees with $(z_1, z_2) \to (z_1, z_1 + z_2)$ on the red rectangle and maps $R_i$ to $Q_i$ such that the uniform measure is preserved.

$$= X_0(z) \cdot \nabla \tilde{f}(|z|) \tag{108}$$
$$= 0. \tag{109}$$

Finally we remark that the flow $\Phi_t$ of the vectorfield $X$ generates a family of functions $h$ as desired, i.e., for all $t$ we have $(\Phi_t)_* Z^{(i)} = Z^{(i)}$ and $\Phi_t$ is clearly nonlinear as it is the identity close to 0 but not globally. $\qquad\square$

The proof of the next lemma is similar but simpler.

**Lemma 8.** *Consider $d = 2$ and $Z^{(0)} = N(0, \mathrm{Id})$ and the non-stochastic hard interventions $\mathrm{do}(Z_1 = 0)$ and $\mathrm{do}(Z_2 = 0)$ resulting in the degenerate Gaussian distributions $Z^{(1)} = N(0, e_2 \otimes e_2)$ and $Z^{(2)} = N(0, e_1 \otimes e_1)$. Then there is a nonlinear $h$ such that $h(Z^{(i)}) \overset{\mathcal{D}}{=} Z^{(i)}$ for all $i$.*

*Proof of Lemma 8.* Pick any smooth divergence free vectorfield $X_0$ with compact support in $[\epsilon, \infty)^2$ for some $\epsilon > 0$. Then the vector field $X = X_0/p_0$ satisfies

$$\mathrm{Div}\, X p_0 = \mathrm{Div}\, X_0 = 0. \tag{110}$$

Thus the flow $\Phi_t$ preserves the distribution of $Z^{(0)}$. But it also preserved the interventional distributions $Z^{(i)}$ for $i = 1, 2$ because $X(z) = 0$ and thus $\Phi_t(z) = z$ for $z$ close to the supports of $Z^{(i)}$. $\qquad\square$

Finally we prove Lemma 9 that shows that we cannot completely drop the assumption on the distribution of the noise variables $\epsilon$.

**Lemma 9.** *Consider $Z^{(0)} = (\epsilon_1, \epsilon_2)^\top$, $Z^{(1)} = (\epsilon_1', \epsilon_2)$, and $Z^{(2)} = (\epsilon_1, \epsilon_2')$ where $\epsilon_1, \epsilon_2 \sim \mathcal{U}([-1, 1])$ and $\epsilon_1', \epsilon_2' \sim \mathcal{U}([-3, 3])$. Also define $\widetilde{Z}^{(0)} = (\epsilon_1, \epsilon_1 + \epsilon_2)$, $\widetilde{Z}^{(1)} = (\epsilon_1', \epsilon_1' + \epsilon_2)$, and $\widetilde{Z}^{(2)} = (\epsilon_1, \epsilon_2')$. Then there is $h$ such that*

$$h(Z^{(i)}) \overset{\mathcal{D}}{=} \widetilde{Z}^{(i)}. \tag{93}$$

*Proof of Lemma 9.* A sketch of the setting can be found in Figure 4. We define $h_0(z) = (z_1, z_1 + z_2)$. If $h(z) = h_0(z)$ for $-1 \leq z_2 \leq 1$ we find $h(Z^{(i)}) \overset{\mathcal{D}}{=} \widetilde{Z}^{(i)}$ for $i = 0$ and $i = 1$. It remains to modify $h_0$ such that $h$ preserves the uniform measure on $[-1, 1] \times [-3, 3]$. We do this in two steps, first we

construct a modification $\widetilde{h}$ of the map $h_0$ such that $[-1, 1] \times [-3, 3]$ is mapped bijectively to itself but $\widetilde{h} = h_0$ for $|z_2| \leq 1$ (so that $\widetilde{h}(Z^{(i)}) \overset{\mathcal{D}}{=} \widetilde{Z}^{(i)}$ for $i = 0$ and $i = 1$ holds) and then we apply a general result to make the map measure preserving. We start with the first step. Let $\psi : \mathbb{R} \to [0, 1]$ be differentiable such that $\psi(t) = 1$ for $-1 \leq z \leq 1$ and $\psi(t) = 0$ for $-5/2 \leq z \leq 5/2$ with $|\psi'(t)| < 1$. Then

$$\tilde{h}(z) = \psi(z_2)z + (1 - \psi(z_2))h_0(z) = \begin{pmatrix} z_1 \\ z_2 + \psi(z_2)z_1 \end{pmatrix} \tag{111}$$

is injective on $[-1, 1] \times [-3, 3]$ (note that the second coordinate is increasing in $z_2$) and maps $[-1, 1] \times [-3, 3]$ bijectively to itself and agrees with $h_0$ for $|z_2| \leq 1$. However, it is not necessarily volume preserving. But applying Moser's theorem (see [56]) to the image of the rectangles $R_1 = [-1, 1] \times [-3, -1]$ and $R_2 = [-1, 1] \times [1, 3]$ which are the quadrilaterals $Q_1 = \tilde{h}(R_1)$ with endpoints $(-1, -3)$, $(1, -3)$, $(1, 0)$, $(-1, -2)$ and $Q_2 = \tilde{h}(R_2)$ with endpoints $(1, 3)$, $(-1, 3)$, $(-1, 0)$, $(1, 2)$ with the same size as $R_1$ and $R_2$ we infer that there is $\varphi$ supported in $Q_1 \cup Q_2$ such that $h = \varphi \circ \tilde{h}$ satisfies $h(Z^{(2)}) \overset{\mathcal{D}}{=} \widetilde{Z}^{(2)} \overset{\mathcal{D}}{=} \mathcal{U}([-1, 1] \times [-3, 3])$. $\qquad\square$

## E   Identifiability and Approximation

In this section, we explain that approximation properties of function classes cannot be leveraged to obtain stronger identifiability results. The general setup is that we have two function classes $\mathcal{G} \subset \mathcal{F}$ and we assume that $\mathcal{G}$ is dense in $\mathcal{F}$ (in the topology of uniform convergence on $\Omega$), i.e., for $f \in \mathcal{F}$ and any $\epsilon > 0$ there is $g \in \mathcal{G}$ such that

$$\sup_{z \in \Omega} |f(z) - g(z)| \leq \epsilon. \tag{112}$$

We now investigate the relationship of identifiability results for mixings in either $\mathcal{G}$ or in $\mathcal{F}$.

While this point is completely general, we will discuss this for concreteness in the context of nonlinear ICA and considering polynomial mixing functions for $\mathcal{G}$. We also assume that $\mathcal{F}$ are all diffeomorphisms (onto their image). Suppose we have latents $Z \sim \mathcal{U}([-1, 1]^d)$ and observe a mixture $X = h(Z)$. We use the shorthand $\Omega = [-1, 1]^d$ from now on. Let us suppose that $h \in \mathcal{G}$. Then the following result holds.

**Lemma 10.** *Suppose $\widetilde{h}(Z) \overset{\mathcal{D}}{=} X \overset{\mathcal{D}}{=} h(Z)$ where $Z \sim \mathcal{U}(\Omega)$, $h, \widetilde{h} \in \mathcal{G}$ and $h$ satisfies an injectivity condition as defined in Assumption 2 in [2]. Then $h$ and $\widetilde{h}$ agree up to permutation of the variables.*

*Proof.* This is a consequence of Theorem 4 in [2]. Essentially, they show in Theorem 1 that $h$ and $\tilde{h}$ agree up to a linear map and then use independence of supports to conclude (in our simpler setting here, one could also resort to the identifiability of linear ICA). $\qquad\square$

The injectivity condition is a technical condition that ensures that the embedding $h$ is sufficiently diverse, but this is not essential for the discussion here. Let us nevertheless mention here that it is not clear whether this condition is sufficiently loose to ensure identifiability in a dense subspace of $\mathcal{F}$ because it requires an output dimension depending on the degree of the polynomials.

We now investigate the implications for $\mathcal{F}$. It is well known that ICA in general nonlinear function is not identifiable, i.e., for any $f \in \mathcal{F}$ there is a $\widetilde{f} \in \mathcal{F}$ such that $f(Z) \overset{\mathcal{D}}{=} \widetilde{f}(Z)$. To find such an $\widetilde{f}$ one use, e.g., the Darmois construction, radius dependent rotations or, more generally, measure preserving transformations $m$ such that $m(Z) \overset{\mathcal{D}}{=} Z$. Then we have the following trivial lemma.

**Lemma 11.** *For every $\epsilon > 0$ there are functions $g, \widetilde{g} \in \mathcal{G}$ such that*

$$\sup_{\Omega} |g - f| < \epsilon, \quad \sup_{\Omega} |\widetilde{g} - \widetilde{f}| < \epsilon \tag{113}$$

*and then the Wasserstein distance satisfies $W_2(f(Z), g(Z)) < \epsilon$, $W_2(\widetilde{f}(Z), \widetilde{g}(Z)) < \epsilon$.*

**Remark 6.** *For the definition and a discussion of the Wasserstein metric we refer to [90]. Alternatively we could add a small amount of noise, i.e. $X = f(Z) + \nu$ and then consider the total variation distance.*

*Proof.* The first part follows directly from the Stone-Weierstrass Theorem which shows that polynomials are dense in the continuous function on compact domains. The second part follows from the coupling $Z \to (f(Z), g(Z))$ and (113). □

The main message of this Lemma is that whenever there are spurious solutions in the larger function class $\mathcal{F}$ we can equally well approximate the ground truth mixing $f$ and the spurious solution $\widetilde{f}$ by functions in $\mathcal{G}$. In this sense, the identifiability of ICA in $\mathcal{G}$ does not provide any guidance to resolve the ambiguity of ICA in $\mathcal{F}$. So identifiability results in $\mathcal{G}$ are not sufficient when there is no reason to believe that the ground truth mixing is exactly in $\mathcal{G}$.

Actually, one could view the approximation capacity of $\mathcal{G}$ also as a slight sign of warning as we will explain now. Suppose the ground truth mixing satisfies $g \in \mathcal{G}$. Since ICA in $\mathcal{F}$ is not identifiable we can find $\widetilde{f} \in \mathcal{F}$ which can be arbitrarily different from $g$ but such that $g(Z) = \widetilde{f}(Z)$. Then we can approximate $\widetilde{f}$ by $\widetilde{g}$ with arbitrarily small error. Thus, we find almost spurious solutions, i.e., $\widetilde{g} \in \mathcal{G}$ such that $W_2(\widetilde{g}(Z), g(Z)) < \epsilon$ for an arbitrary $\epsilon > 0$ but $\widetilde{g}$ and $g$ correspond to very different data representations, in this sense the identifiability result is not robust.

When only considering the smaller space $\mathcal{G}$ this problem can be resolved by considering norms or seminorms on $\mathcal{G}$, for polynomials a natural choice is the degree of the polynomial. Indeed, suppose that we observe data generated using a mixing $g \in \mathcal{G}$. Then there might be spurious solutions $\widetilde{f} \in \mathcal{F}$ which can be approximated by $\widetilde{g} \in \mathcal{G}$ but this approximation will generically have a larger norm (or degree for polynomials) than $g$. Therefore, the minimum description length principle [69] favors $g$ over $\widetilde{g}$.

This reasoning can only be extended to the larger class $\mathcal{F}$ if the ground truth mixing $f$ (for the relevant applications) can be better approximated (i.e., with smaller norm) by functions in $\mathcal{G}$ than all spurious solutions $\widetilde{f}$. This is hard to verify in practice and difficult to formalize theoretically. Nevertheless, this motivates to look for identifiability results for function classes that are known to be useful representation learners and used in practice, in particular neural nets. We emphasize that alternatively one can also directly consider norms on the larger space $\mathcal{F}$ penalizing, e.g., derivative norms, to perform model selection.

# F   Additional details on experimental methodology

In this appendix, we give more details about our experimental methodology. First, we derive the log-odds and prove Lemma 1 in Appendix F.1. Then, we use it to design our model and theoretically justify our contrastive approach in Appendix F.2, by connecting it to the Bayes optimal classifier. Then, we describe the NOTEARS regularizer in Appendix F.3, describe the limitations of the contrastive approach in Appendix G.1 and finally, in Appendix F.4, we describe the ingredients needed for an approach via Variational Autoencoders, the difficulties involved and how to potentially bypass them.

## F.1   Proof of Lemma 1

**Lemma 1.** *When we have perfect interventions, the log-odds of a sample $x$ coming from $X^{(i)}$ over coming from $X^{(0)}$ is given by*

$$\ln p_X^{(i)}(x) - \ln p_X^{(0)}(x) = c_i - \frac{1}{2}\lambda_i^2(((f^{-1}(x))_{t_i})^2 + \eta^{(i)}\lambda_i \cdot (f^{-1}(x))_{t_i}) + \frac{1}{2}\langle f^{-1}(x), s^{(i)}\rangle^2 \quad (6)$$

*for a constant $c_i$ independent of $x$.*

*Proof.* We will write the log likelihood of $X^{(i)} = f(Z^{(i)})$ using standard change of variables. As we elaborate in Appendix A.1, we have

$$Z^{(i)} = (B^{(i)})^{-1}\epsilon + \mu^{(i)} \text{ with } B^{(i)} = (D^{(i)})^{-1/2}(I - A^{(i)}), \mu^{(i)} = \eta^{(i)}(B^{(i)})^{-1}e_{t_i} \quad (114)$$

Also, denote by $\Theta^{(i)} = (B^{(i)})^\top B^{(i)}$ the precision matrix of $Z^{(i)}$ (see Appendix A.1 for full derivation). By change of variables,

$$\ln p_X^{(i)}(x) = \ln |J_{f^{-1}}| + \ln p_Z^{(i)}(f^{-1}(x))$$

$$= \ln |J_{f^{-1}}| - \frac{n}{2} \ln(2\pi) - \frac{1}{2} \ln |\Sigma^{(i)}| - \frac{1}{2}(f^{-1}(x) - \mu^{(i)})^T \Theta^{(i)}(f^{-1}(x) - \mu^{(i)})$$

where $J_{f^{-1}}$ denotes the Jacobian matrix of partial derivatives. Let $s^{(1)}, s^{(2)}, \ldots, s^{(d)}$ denote the rows of $B^{(0)}$. We then compute the log-odds with respect to $X^{(0)}$, the base distribution without interventions (and where $\mu^{(0)} = 0$) to get

$$\ln p_X^{(i)}(x) - \ln p_X^{(0)}(x)$$
$$= \left( -\frac{1}{2} \ln \frac{|\Sigma^{(i)}|}{|\Sigma^{(0)}|} - \frac{1}{2}(\mu^{(i)})^\top \Theta^{(i)} \mu^{(i)} \right) - \frac{1}{2}(f^{-1}(x))^\top (\Theta^{(i)} - \Theta^{(0)})(f^{-1}(x)) + (f^{-1}(x))^\top \Theta^{(i)} \mu^{(i)}$$
$$= c_i - \frac{1}{2}(f^{-1}(x))^\top (r^{(i)} \otimes r^{(i)} - s^{(i)} \otimes s^{(i)})(f^{-1}(x)) + \eta^{(i)} \cdot (f^{-1}(x))^\top (B^{(i)})^\top e_{t_i}$$
$$= c_i - \frac{1}{2}(f^{-1}(x))^\top (\lambda_i^2 e_{t_i} e_{t_i}^\top - (s^{(i)})(s^{(i)})^\top)(f^{-1}(x)) + \eta^{(i)} \lambda_i \cdot (f^{-1}(x))^\top e_{t_i}$$
$$= c_i - \frac{1}{2}(f^{-1}(x))^\top (\lambda_i^2 e_{t_i} e_{t_i}^\top - (s^{(i)})(s^{(i)})^\top)(f^{-1}(x)) + \eta^{(i)} \lambda_i \cdot (f^{-1}(x))_{t_i}$$
$$= c_i - \frac{1}{2}\lambda_i^2(((f^{-1}(x))_{t_i})^2 + \eta^{(i)} \lambda_i \cdot (f^{-1}(x))_{t_i}) + \frac{1}{2}\langle f^{-1}(x), s^{(i)} \rangle^2$$

for a constant $c_i$ independent of $z$. For the second equality, we used Lemma 3. $\qquad\square$

## F.2 A theoretical justification for the log-odds model

In this section, we provide more details on our precise modeling choices for the contrastive approach, and show theoretically why it encourages the model to learn the true underlying parameters.

Recall that, motivated by Lemma 1, we model the log-odds as

$$g_i(x, \alpha_i, \beta_i, \gamma_i, w^{(i)}, \theta) = \alpha_i - \beta_i h_{t_i}^2(x, \theta) + \gamma_i h_{t_i}(x, \theta) + \langle h(x, \theta), w^{(i)} \rangle^2 \tag{115}$$

where $h(\cdot, \theta)$ denotes a neural net parametrized by $\theta$, parameters $w^{(i)}$ are the rows of a matrix $W$ and $\alpha_i, \beta_i, \gamma_i$ are learnable parameters. Moreover, we have $g_0(x) = 0$ as we compute the log-odds with respect to $X^{(0)}$. Therefore, the cross entropy loss given by

$$\mathcal{L}_{\text{CE}}^{(i)} = -\mathbb{E}_{j \sim \mathcal{U}(\{0,i\})} \mathbb{E}_{x \sim X^{(j)}} \ln \left( \frac{e^{\mathbf{1}_{j=i} g_i(x)}}{e^{g_i(x)} + 1} \right). \tag{116}$$

Let $C(x) \in \{0, 1\}$ denote the label of the datapoint $x$ indicating whether it was an observational datapoint or an interventional datapoint respectively. By using the cross entropy loss, we are treating the model outputs as logits, therefore we can write down the posterior probability distribution using a logistic regression model as

$$\Pr[C(x) = 1|x] = \frac{\exp(g_i(x, \alpha_i, \beta_i, \gamma_i, w^{(i)}, \theta))}{1 + \exp(g_i(x, \alpha_i, \beta_i, \gamma_i, w^{(i)}, \theta))} \tag{117}$$

Moreover, by using a sufficiently wide or deep neural network with universal approximation capacity and sufficiently many samples, our model will learn the Bayes optimal classifier, as given by Lemma 1. Therefore, we can equate the probabilities to arrive at

$$\frac{\exp(g_i(x, \alpha_i, \beta_i, \gamma_i, w^{(i)}, \theta))}{1 + \exp(g_i(x, \alpha_i, \beta_i, \gamma_i, w^{(i)}, \theta))} = \frac{\exp(\ln p_X^{(i)}(x) - \ln p_X^{(0)}(x))}{1 + \exp(\ln p_X^{(i)}(x) - \ln p_X^{(0)}(x))} \tag{118}$$

implying

$$\alpha_i - \beta_i h_{t_i}^2(x, \theta) + \gamma_i h_{t_i}(x, \theta) + \langle h(x, \theta), w^{(i)} \rangle^2 \tag{119}$$
$$= c_i - \frac{1}{2}\lambda_i^2(((f^{-1}(x))_{t_i})^2 + \eta^{(i)} \lambda_i \cdot (f^{-1}(x))_{t_i}) + \frac{1}{2}\langle f^{-1}(x), s^{(i)} \rangle^2 \tag{120}$$

This suggests that in optimal settings, our model will learn (up to scaling)

$$\alpha_i = c_i, \quad \beta_i = \frac{1}{2}\lambda_i^2, \quad \gamma_i = \eta^{(i)} \lambda_i, \quad w^{(i)} = s^{(i)}, \quad h(x, \theta) = f^{-1}(x) \tag{121}$$

thereby inverting the nonlinearity and learning the underlying parameters.

## F.3 NOTEARS [102]

In our algorithm, using the parameters $w^{(i)}$ as rows, we learn the matrix $W$, which we saw in optimal settings will be $B = D^{-1/2}(\text{Id} - A)$. Note that the off-diagonal entries of $B$ form a DAG. Therefore, the graph encoded by $W_0$, which is defined to be $W$ with the main diagonal zeroed out, must also be a DAG.

However, in our algorithm, we don't explicitly enforce this acyclicity. There are two ways to get around this. One way is to assume a default causal ordering on the $Z_i$s, which is feasible as we don't directly observe $Z$. Then, we can simply enforce $W_0$ to be triangular and train via standard gradient descent and $W_0$ is guaranteed to be a DAG. However, this approach doesn't work because the interventional datasets are given in arbitrary order and so we are not able to match the datasets to the vertices in the correct order. So this merely defers the issue.

Instead, the other approach that we take in this work is to regularize the learnt $W_0$ to model a directed acyclic graph. Learning an underlying causal graph from data is a decades old problem that has been widely studied in the causal inference literature, see e.g. [12, 102, 64, 80, 4] and references therein. In particular, the work [102] proposed an analytic expression to measure the DAGness of the causal graph, thereby making the problem continuous and more efficient to optimize over.

**Lemma 12** ([102]). *A matrix $W \in \mathbb{R}^{d \times d}$ is a DAG if and only if*

$$\mathcal{R} := \text{tr} \exp(W \circ W) - d = 0 \tag{122}$$

*where $\circ$ is the matrix Hadamard product and $\exp$ is the matrix exponential. Moreover, $\mathcal{R}$ has gradient*

$$\nabla \mathcal{R} = \exp(W \circ W)^\top \circ 2W \tag{123}$$

Therefore, we add $\mathcal{R}_{NOTEARS}(W) = \text{tr} \exp(W_0 \circ W_0) - d$ as a regularization to our loss function. As in prior works, we could also consider the augmented Lagrangian formulation however it leads to additional training complexity. There have been several follow-up works to NOTEARS such as [98] (see [4] for an overview) that suggest different regularization schemes. We leave exploring such alternatives to future work.

## F.4 A Variational Autoencoder approach

In this section, we describe the technical details involved with adapting Variational Autoencoders (VAEs) [39, 68] for our setting. We highlight some difficulties that we will encounter and also suggest possible ways to work around them.

Indeed, most earlier approaches for causal representation learning have relied on maximum likelihood estimation via variational inference. In particular, they have relied on autoencoders or variational autoencoders [36, 2]. In our setting, VAEs are a viable approach. More concretely, we can encode the distribution $X^{(0)}$ to $\epsilon$, then apply the $B^{(i)}$ parameter matrix before finally decoding to $X^{(i)}$. To handle the fact that we don't have paired interventional data as in [6], we could potentially modify the Evidence Lower Bound (ELBO) to include some divergence measure between the predicted and measured interventional data. Finally, this can be trained end-to-end as in traditional VAEs. We now describe this in more detail.

We will use VAEs to encode the observational distribution $X^{(0)}$ to $\epsilon$, so the encoder will formally model $B^{(0)} \circ (f^{-1}(.) - \mu^{(i)})$ where we use the notation from Appendix A.1. Note here that we cannot directly design the encoder to map $X$ to $Z$ because we do not know the prior distribution on $Z$. Indeed, the objective is to learn it.

Let $I$ denote the set of intervention targets. Assume the targets are chosen uniformly at random, which can be done in practice by subsampling each interventional distribution to have the same size. Suppose we had paired counterfactual data $\mathcal{D} = \cup_{i \in I} \mathcal{D}_i$ of paired interventional datasets, i.e. $\mathcal{D}_i$ consists of pairs $(X^{(0)}, X^{(i)})$ with corresponding probability density denoted $p^{(i)}(x^{(0)}, x^{(i)})$. Following [39, 101], define an amortized inference distribution as $q(\epsilon|x^{(0)})$. Then, we can bound the expected log-likelihood as

$$\mathbb{E}_{\mathcal{D}}[\ln p^{(i)}(x^{(0)}, x^{(i)})] = \mathbb{E}_{i \sim \mathcal{U}(I)} \mathbb{E}_{\mathcal{D}_i}[\ln p^{(i)}(x^{(0)}, x^{(i)})] \tag{124}$$

$$= \mathbb{E}_{i \sim \mathcal{U}(I)} \mathbb{E}_{\mathcal{D}_i} \ln \int_\epsilon p^{(i)}(x^{(0)}, x^{(i)}, \epsilon) \, d\epsilon \tag{125}$$

$$= \mathbb{E}_{i \sim \mathcal{U}(I)} \mathbb{E}_{\mathcal{D}_i} \ln \int_\epsilon \frac{p^{(i)}(x^{(0)}, x^{(i)}, \epsilon) q(\epsilon|x^{(0)})}{q(\epsilon|x^{(0)})} \, d\epsilon \tag{126}$$

$$= \mathbb{E}_{i \sim \mathcal{U}(I)} \mathbb{E}_{\mathcal{D}_i} \ln \mathbb{E}_{\epsilon \sim q(\epsilon|x^{(0)})} \frac{p^{(i)}(x^{(0)}, x^{(i)}, \epsilon)}{q(\epsilon|x^{(0)})} \tag{127}$$

$$\geq \mathbb{E}_{i \sim \mathcal{U}(I)} \mathbb{E}_{\mathcal{D}_i} \mathbb{E}_{\epsilon \sim q(x^{(0)})} \ln \frac{p^{(i)}(x^{(0)}, x^{(i)}, \epsilon)}{q(\epsilon|x^{(0)})} \qquad \text{(Jensen's inequality)} \tag{128}$$

$$= \mathbb{E}_{i \sim \mathcal{U}(I)} \mathbb{E}_{\mathcal{D}_i} \mathbb{E}_{\epsilon \sim q(x^{(0)})} \ln \frac{p^{(i)}(x^{(0)}, x^{(i)}|\epsilon) p(\epsilon)}{q(\epsilon|x^{(0)})} \tag{129}$$

$$= \mathbb{E}_{i \sim \mathcal{U}(I)} \mathbb{E}_{\mathcal{D}_i} \left( \mathbb{E}_{\epsilon \sim q(x^{(0)})} \ln p^{(i)}(x^{(0)}, x^{(i)}|\epsilon) - \mathbb{E}_{\epsilon \sim q(x^{(0)})} \ln \frac{q(\epsilon|x^{(0)})}{p(\epsilon)} \right) \tag{130}$$

$$= \mathbb{E}_{i \sim \mathcal{U}(I)} \mathbb{E}_{(x^{(0)}, x^{(i)}) \sim \mathcal{D}_i} \left[ \mathbb{E}_{\epsilon \sim q(\epsilon|x^{(0)})} [\ln p^{(i)}(x^{(0)}, x^{(i)}|\epsilon)] \right. \tag{131}$$

$$\left. - \mathrm{KL}(q(\epsilon|x^{(0)})||N(0, I)) \right] \tag{132}$$

where the last term is the standard Evidence Lower Bound (ELBO). This can then be trained end-to-end via the reparameterization trick. A similar approach was taken in [6] who had access to paired counterfactual data.

However, we do not have access to such paired interventional data, but instead we only observe the marginal distributions $X^{(i)}$. To this end, conditioned on $\epsilon$, we can split the term $\ln p^{(i)}(x^{(0)}, x^{(i)}|\epsilon) = \ln \widetilde{p}^{(0)}(x^{(0)}|\epsilon) + \ln \widetilde{p}^{(i)}(x^{(i)}|\epsilon)$ where $\widetilde{p}^{(i)}$ denotes the density of the intervened marginal of $p^{(i)}$, which corresponds to using the interventional decoder $f \circ ((B^{(i)})^{-1}(.) + \mu^{(i)})$. Nevertheless, $\mathbb{E}_{\epsilon \sim q(x^{(0)})}[\ln \widetilde{p}^{(i)}(x^{(i)}|\epsilon)]$ is still not tractable in unpaired settings. As a heuristic, we could consider a modified ELBO by modifying this term to measure some sort of divergence metric between the true interventional data $X^{(i)}$ and the generated interventional data $\widetilde{p}^{(i)}(x^{(i)}|\epsilon)$, similar to [101]. We leave it for future work to explore this direction.

# G  Additional experimental results

In this section, we provide additional experimental results and discussions to explore the effect of shift strength of the interventions, data scale, and noise distribution.

## G.1  Limitations of the contrastive approach

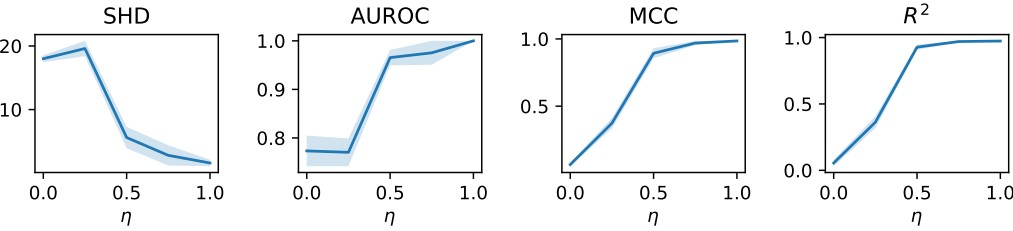

Figure 5: Dependence of the performance metrics on the shift $\eta$. Other settings are as for nonlinear mixing functions (see Table 7) with $\mathrm{ER}(10, 2)$ graphs and $d' = 100$.

As mentioned in the main part of the paper, our contrastive algorithm struggles to recover the ground truth latent variables when considering interventions without shifts. We illustrate the dependence on the shift strength in Figure 5. In this section, we provide some evidence why this is the case. Apart

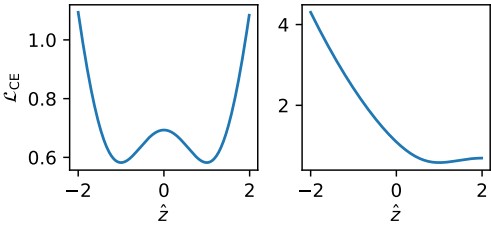

Figure 6: Cross entropy loss for $a = 1$, $c = 0$, $z_0 = 1$ and $b = 0$ (left), $b = 2$ (right) as a function of the estimated latent variable $\hat{z}$.

from setting the stage for future works, this also enables practitioners to be aware of potential pitfalls when applying our techniques.

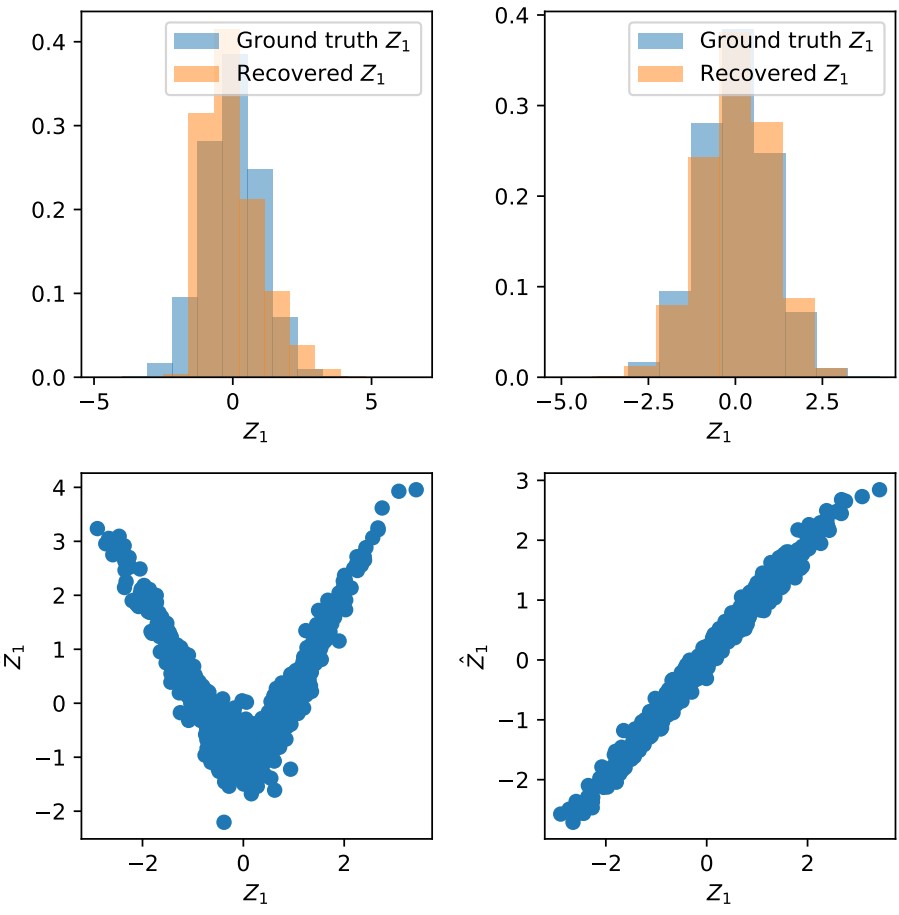

Figure 7: Density of standardized ground truth latents and recovered latents for $\eta = 0$ (top left) and $\eta = 1$ (top right), scatter plot of the ground truth latent variables $Z_1$ and recovered latent variables $\hat{Z}_1$ for $\eta = 0$ (bottom left) and $\eta = 1$ (bottom right). Results shown for $\mathrm{ER}(10, 2)$ graphs and $d' = 100$, all further parameters as in Table 7.

We suspect that the main reason for this behavior is the non-convexity of the parametric output layer. Note that this is very different from the well known non-convexity of (overparametrized) neural networks. While neural networks parametrize non-convex functions their final layer is typically a convex optimization problem, e.g., a least squares regression or a logistic regression on the features.

Moreover, it is well understood that convergence to a global minimizer using gradient descent is possible under suitable conditions, see, e.g., [47].

This is different for our quadratic log-odds expression where gradient descent is not sufficient to find the global minimizer. To illustrate this we consider the case of a single latent, i.e., $d = 1$, then the ground truth parametric form of the log-odds in 6 can be expressed as

$$\ln p_X^{(1)} - \ln p_X^{(0)}(x) = g(x) = h(f^{-1}(x)) = a(f^{-1}(x) - b)^2 + c \tag{133}$$

for some constants $a$, $b$, $c$ and $b = 0$ if $\eta^{(1)} = 0$, i.e., there is no shift. We moreover assume, for the sake of argument, that the final layer implementing $h(z) = a(z - b)^2 + c$ is fixed to the ground truth parametric form of the log-odds (then there is also no scaling ambiguity left). Now, let us fix a point $x_0$ and define $z_0 = f^{-1}(x_0)$ and

$$p = \mathbb{P}(i = 1 | X = x_0) = \frac{p_X^{(1)}(x_0)}{p_X^{(1)}(x_0) + p_X^{(0)}(x_0)} = \frac{e^{g(x_0)}}{1 + e^{g(x_0)}}. \tag{134}$$

Then the (sample conditional) cross entropy loss as a function of $\hat{z} = \hat{f}^{-1}(x_0)$ for our learned function $\hat{f}$ is given by

$$\mathcal{L}_{\text{CE}} = -p \ln \left( \frac{e^{h(\hat{z})}}{1 + e^{h(\hat{z})}} \right) - (1 - p) \ln \left( \frac{1}{1 + e^{h(\hat{z})}} \right) = -p h(\hat{z}) + \ln(1 + e^{h(\hat{z})}) \tag{135}$$

We here drop the intervention index, since we focus on a one dimensional illustration. We plot two examples of such loss functions for $b = 0$ and $b \neq 0$ in Figure 6.

To verify that this is indeed the reason for our failure to recover the ground truth latent variables, we investigate the relation between estimated latent variables and ground truth latent variables. The result can be found in Figure 7 and, as we see, when $\eta = 0$, the distribution of the recovered latents is skewed. Moreover, we find that we essentially recover the latent variable up to a sign, i.e., $\hat{Z} = |Z|$ (there is a small offset because we enforce $\hat{Z}$ to be centered). Note that this explains the tiny MCC scores in Table 1, since $\mathbb{E}(Z|Z|) = 0$ for symmetric distributions. This observation might also be a potential starting point to improve our algorithm. For non-zero shifts we recover the latent variables almost perfectly.

### G.2 Varsortability and data scale

An important observation in causal discovery was that many benchmark settings exhibit specific correlation structures that can be exploited to infer causal directions. The most prominent example is varsortability [66] which measures how well the causal order agrees with the order induced by the magnitude of the variance. This is a measure between 0 and 1 and for values close to 0 and 1 the variances contain a lot of information about the causal order while close to 0.5 this is not the case. It was shown in [66] that typical parameter choices exhibit high varsortability $\approx 1$ which can be exploited by trivial algorithms. A related notion is $R^2$-varsortability [67] which is a similar notion where the variance of the variable is replaced by the residual variance after standardizing the variables and then regressing out all other variables, i.e., the unexplained variance.

In our setting the varsortability of the latent variables $Z$ is quite high (see [66] for the full definition), e.g., for $d = 10$ and $k = 2$ average varsortability is $0.92 \pm 0.02$. Note that it is not obvious how this can be exploited algorithmically. Indeed, we only observe a mixture $X = f(Z)$ and the scale of $Y$ is not identifiable. Nevertheless, to rule out that we implicitly use the scale of the variables we ran the same experiments except that we standardized the latent variables $Z$ of the observable distribution (and apply the same scaling to the interventional distributions). The results can be found in Table 4 and are roughly similar to the result in Table 2 for the same settings without standardization. Note that $R^2$-sortability is not substantially different from .5 (roughly $0.4$) in our settings, thus the $R^2$-sortability cannot be exploited to infer the causal structure, even when observing $Z$.

### G.3 Effect of noise distribution

Our theoretical results only cover the setting of Gaussian SCMs and we even show that identifiability does not necessarily hold for uniformly distributed noise (see Appendix D.2). Nevertheless, we can

Table 4: Results for nonlinear synthetic data with $n = 10000$ and standardized latent variables $Z$ (up to standardization the setting is the same as in Table 2 in the paper).

| Setting | Method | SHD $\downarrow$ | AUROC $\uparrow$ | MCC $\uparrow$ | $R^2 \uparrow$ |
|---|---|---|---|---|---|
| ER(5, 2), $d' = 20$ | Contrastive Learning | $2.6 \pm 0.4$ | $0.92 \pm 0.02$ | $0.96 \pm 0.01$ | $0.93 \pm 0.01$ |
| | VAE | $10.0 \pm 0.0$ | $0.50 \pm 0.00$ | $0.11 \pm 0.01$ | $0.04 \pm 0.01$ |
| ER(5, 2), $d' = 100$ | Contrastive Learning | $1.4 \pm 0.4$ | $0.98 \pm 0.02$ | $0.98 \pm 0.00$ | $0.97 \pm 0.01$ |
| | VAE | $10.0 \pm 0.0$ | $0.50 \pm 0.00$ | $0.14 \pm 0.03$ | $0.10 \pm 0.04$ |
| ER(10, 2), $d' = 20$ | Contrastive Learning | $9.6 \pm 2.1$ | $0.90 \pm 0.03$ | $0.90 \pm 0.01$ | $0.84 \pm 0.01$ |
| | VAE | $18.6 \pm 0.9$ | $0.50 \pm 0.00$ | $0.27 \pm 0.04$ | $0.29 \pm 0.04$ |
| ER(10, 2), $d' = 100$ | Contrastive Learning | $2.8 \pm 1.4$ | $0.99 \pm 0.01$ | $0.98 \pm 0.00$ | $0.96 \pm 0.00$ |
| | VAE | $18.6 \pm 0.9$ | $0.50 \pm 0.00$ | $0.14 \pm 0.03$ | $0.11 \pm 0.04$ |

Table 5: Noise dependence of contrastive algorithm for ER(5, 2), $d' = 20$, $n = 10000$ and nonlinear mixing (same setting as first row of Table 2 in the paper).

| Noise distribution | SHD $\downarrow$ | AUROC $\uparrow$ | MCC $\uparrow$ | $R^2 \uparrow$ |
|---|---|---|---|---|
| Gaussian | $1.8 \pm 0.5$ | $0.97 \pm 0.01$ | $0.97 \pm 0.00$ | $0.96 \pm 0.00$ |
| Laplace | $3.0 \pm 0.6$ | $0.89 \pm 0.03$ | $0.93 \pm 0.01$ | $0.89 \pm 0.01$ |
| Gumbel | $3.0 \pm 0.9$ | $0.92 \pm 0.02$ | $0.94 \pm 0.01$ | $0.91 \pm 0.01$ |
| Uniform | $3.4 \pm 0.4$ | $0.87 \pm 0.03$ | $0.96 \pm 0.01$ | $0.93 \pm 0.01$ |
| Exponential | $3.8 \pm 0.5$ | $0.86 \pm 0.02$ | $0.89 \pm 0.02$ | $0.86 \pm 0.02$ |

evaluate the performance of our algorithm for linear SCMs with non-Gaussian noise distribution. We rerun one of the settings in Table 2 for different noise distributions. The results can be found in Table 5, and they show a slightly worse but still reasonable performance than for Gaussian data. This is in line with the observation that using a Gaussian likelihood for non-Gaussian data gives good results in, e.g., causal discovery.

## H  Hyperparameters, architectures and further additional details on experiments

In this section, we give additional details on the experiments. We use PyTorch [60] for all our experiments.

**Data generation**   To generate the latent DAG and sample the latent variables we use the sempler package [21]. We always sample $\mathrm{ER}(d, k)$ graphs and the non-zero weights are sampled from the distribution $w_{ij} \sim \mathcal{U}(\pm[0.25, 1.0])$. To generate linear synthetic data, we sample all entries of the mixing matrix i.i.d. from a standard Gaussian distribution. For non-linear synthetic data $f$ is given by the architecture in Table 9 with weights sampled from $\mathcal{U}([-\frac{1}{\sqrt{in_{feat}}}, \frac{1}{\sqrt{in_{feat}}}])$. For image data, similar to [2], the latents $Z$ are the coordinates of balls in an image (but we sample them differently as per our setting) and a $64 \times 64 \times 3$ RGB image is rendered using Pygame [77] which encompasses the nonlinearity. Other parameter choices such as dimensions, DAGs, variance parameters, and shifts are already outlined in Section 6 but for clarity we also outline them in Tables 6, 7, and 8.

**Models**   For synthetic data, the model architecture for $h(x, \theta)$ is a one hidden-layer MLP with 512 hidden units and LeakyReLU activation functions. For the parametric final layer, we decompose the matrix $W$ as $W = D(\mathrm{Id} - A)$ where $D$ is a diagonal matrix and $A$ is a matrix with a masked diagonal and they are both learned. This factorization shall improve stability. We initialize $A = 0$ and $D = \mathrm{Id}$. Also, the sparsity penalty and the DAGness penalty are only applied to $A$. For the baseline VAE we use the same architecture for both encoder and decoder. The code for the linear baseline is from [84].

Table 6: Parameters used for linear synthetic data.

| | |
|---|---|
| $d$ | 5 |
| $d'$ | 10 |
| $k$ | 3/2 |
| $n$ | $\{2500, 5000, 10000, 25000, 50000\}$ |
| $\sigma^2_{\mathrm{obs}}$ | $\mathcal{U}([2,4])$ |
| $\sigma^2_{\mathrm{int}}$ | $\mathcal{U}([6,8])$ |
| $\eta^{(i)}$ | 0 |
| runs | 5 |
| mixing | linear |

Table 7: Parameters used for nonlinear synthetic data.

| | |
|---|---|
| $d$ | $\{5, 10\}$ |
| $d'$ | $\{20, 100\}$ |
| $k$ | $\{1, 2\}$ |
| $n$ | 10000 |
| $\sigma^2_{\mathrm{obs}}$ | $\mathcal{U}([1,2])$ |
| $\sigma^2_{\mathrm{int}}$ | $\mathcal{U}([1,2])$ |
| $\eta^{(i)}$ | $\mathcal{U}(\pm[1,2])$ |
| runs | 5 |
| mixing | 3 layer mlp |

Table 8: Parameters used for image data.

| | |
|---|---|
| $d$ | $\{4, 6\}$ |
| $d'$ | $3 \cdot 64^2$ |
| $k$ | $\{1, 2\}$ |
| $n$ | 25000 |
| $\sigma^2_{\mathrm{obs}}$ | $\mathcal{U}([0.01, 0.01])$ |
| $\sigma^2_{\mathrm{int}}$ | $\mathcal{U}([0.01, 0.02])$ |
| $\eta^{(i)}$ | $\mathcal{U}(\pm[0.1, 0.2])$ |
| runs | 5 |
| mixing | image rendering |

Table 9: Synthetic data generation

| Architecture | Layer Sequence |
|---|---|
| *MLP Embedding* | Input: $z \in \mathbb{R}^d$ |
| | FC 512, LeakyReLU(0.2) |
| | FC 512, LeakyReLU(0.2) |
| | FC 512, LeakyReLU(0.2) |
| | FC $d'$ $\longleftarrow$ Output $x$ |

For image data, the model architectures for $h(x, \theta)$ are small convolutional networks. For the contrastive approach the architecture can be found in Table 10. For the VAE model, we use the encoder architecture as in Table 11 and the decoder architecture as in Table 12 respectively.

Table 10: Contrastive model architecture for image data.

| Architecture | Layer Sequence |
|---|---|
| *Conv Embedding* | Input: $x \in \mathbb{R}^{64 \times 64 \times 3}$ |
| | Conv (3, 10, kernel size = 5, stride = 3), ReLU() |
| | MaxPool (kernel size = 2, stride = 2) |
| | FC 64, LeakyReLU() |
| | FC $d$ $\longleftarrow$ Output $\hat{z}$ |

**Training** For training we use the hyperparameters outlined in Table 13. We use a 80–20 split for train and validation/test set respectively. After subsampling each dataset to the same size, we copy the observation samples for each interventional dataset in order to have an equal number of observational and interventional samples so they can be naturally paired during the contrastive learning. We select the model with the smallest validation loss on the validation set, where we only use the cross entropy loss for validation. For the VAE baseline we use the standard VAE validation loss for model selection.

Table 11: VAE Encoder architecture for image data

| Architecture | Layer Sequence |
|---|---|
| *Conv Encoder* | Input: $x \in \mathbb{R}^{64 \times 64 \times 3}$ |
| | Conv(3, 16, kernel size = 4, stride = 2, padding = 1), LeakyReLU()
Conv(16, 32, kernel size = 4, stride = 2, padding = 1), LeakyReLU()
Conv(32, 32, kernel size = 4, stride = 2, padding = 1), LeakyReLU()
Conv(32, 64, kernel size = 4, stride = 2, padding = 1), LeakyReLU()
FC $2d$ $\longleftarrow$ Output (mean, logvar) |

Table 12: VAE Decoder architecture for image data

| Architecture | Layer Sequence |
|---|---|
| *Deconv Decoder* | Input: $\hat{z} \in \mathbb{R}^d$ |
| | FC $(3 \times 3) \times d$
DeConv $(d, 32,$ kernel size = 4, stride = 2, padding = 0), LeakyReLU()
DeConv $(32, 16,$ kernel size = 4, stride = 2, padding = 1), LeakyReLU()
DeConv $(16, 8,$ kernel size = 4, stride = 2, padding = 1), LeakyReLU()
DeConv $(8, 3,$ kernel size = 4, stride = 2, padding = 1), LeakyReLU()
Sigmoid() $\longleftarrow$ Output $\hat{x}$ |

Table 13: Hyperparameters used for training.

| | |
|---|---|
| $\tau_1$ (see Eq. (9)) | $10^{-5}$ |
| $\tau_2$ (see Eq. (9)) | $10^{-4}$ |
| learning rate | $5 \cdot 10^{-4}$ |
| batch size | 512 |
| epochs | 200 (synthetic data), 100 (image data) |
| optimizer | Adam [38] |
| learning rate scheduler | CosineAnnealing [51] |

**Compute and runtime** The experiments using synthetic data were run on 2 CPUs with 16Gb RAM on a compute cluster. For image data we added a GPU to increase the speed. The runtime per epoch for the synthetic data and 10000 samples was roughly 10s. The total run-time of the reported experiments was about 100h (i.e. 200 CPU hours) and 20h on a GPU, but hyperparameter selection and preliminary experiments required about 10 times more CPU compute.

**Mean Correlation Coefficient (MCC)** Mean Correlation Coefficient (MCC) is a metric that has been utilized in prior works [37, 41] to quantify identifiability. MCC measures linear correlations up to permutation of the components. To compute the best permutation, a linear sum assignment problem is solved and finally, the correlation coefficients are computed and averaged over. A high MCC indicates that the true latents have been recovered. In this work, we compute the transformation using half of the samples and then report the MCC using the other half, i.e. the out-of-distribution samples.

**Further experimental attempts and dead Ends** We will briefly summarize a few additional settings we tried in our experiments.

- We fixed $D = \mathrm{Id}$ in the parametrization of $W$. This fixes the scaling of the latent variables in some non-trivial way. This resulted in very similar results.

- We tried to combine the contrastive algorithm with a VAE approach. Since the ground truth latent variables $Z$ are highly correlated, they are not suitable for an autoencoder that typically assumes a factorizing prior. Therefore, we map the estimated latent variables for the observational distribution to the noise distribution $\epsilon$ which factorizes. Note that for the ground truth latent variables the relation $\epsilon = B^{(0)} Z^{(0)}$ holds so we consider $\hat{\epsilon} = W \hat{Z}$. We

use $\hat{\epsilon}$ as the latent space of an autoencoder on which we then stack a decoder. We train using the observational samples and the ELBO for the VAE part and with the contrastive loss (and interventional and observational samples) for the $\hat{Z}$ embedding. This resulted in a slightly worse recovery of the latent variables (as measured by the MCC score) but much worse recovery of the graph. We suspect that this originates from the fact that the matrix $W$ is used both in the parametrization of the log-odds and to map $\hat{Z}$ to $\epsilon$ which might make the learning more error-prone. Note that this is different from the suggestion in Section F.4.

- Choosing a larger learning rate for the parametric part than for the non-parametric part seemed to help initially, but using the same sufficiently small learning rate for the entire model was more robust.

- We tried larger architectures for both synthetic and image data. They turned out to be more difficult to train without offering any benefit.

- For image data we often observed overfitting when using smaller sample sizes. The overfitting persisted even when we used a pre-trained ResNet [24] with frozen layers. We suspect that for image data the performance (in particular the sample complexity) can be further improved by carefully tuning regularization and model size.

