# OpenReview forum: "Learning Linear Causal Representations from Interventions under General Nonlinear Mixing"
_NeurIPS.cc/2023/Conference — NeurIPS 2023 oral_

### Official Review · Reviewer_RqDB · 2023-07-01

**Soundness:** 2 fair
**Presentation:** 3 good
**Contribution:** 2 fair
**Rating:** 7
**Confidence:** 3

**Summary:**

The authors study the problem of learning latent causal models from interventional data. The problem was formulated under linear or polynomial mixing in previous works and this work considers a more general setting of nonlinear mixing. The main contributions are the identifiability results of the latent causal model and a contrastive algorithm that identifies the latent model.

**Strengths:**

1. The extension to nonlinear mixing is an important and challenging problem.

2. Using the contrastive algorithm to identify the model parameters is novel and sound.

**Weaknesses:**

1. Single-node interventions are considered, while more nodes can be intervened in each environment in practice. For theoretical results, I think focusing on single-node intervention is fine, but it should be discussed whether the results have the potential to be generalized.

2. It was not mentioned whether the latent dimension $d$ is identifiable in the nonlinear setting. For all the identifiability results in the paper, the considered $\widetilde{Z}^{(i)}$ is assumed to have the same dimension as the true latent variable $Z^{(i)}$ (i.e., $d$). Note that $d$ is identifiable and it equals to the rank of the precision matrix of X in the linear setting. (Section 3.1 [1]).

This issue occurs in the experiments as well.  $d$ is provided to the contrastive method. But it is often not possible to know $d$ in practical settings.

[1] C. Squires, A. Seigal, S. Bhate, and C. Uhler. Linear causal disentanglement via interventions, 2023.

Additional comment: I think it would be helpful to provide a concrete toy example to demonstrate the identifiability since the proof intuition is a bit vague.






**Questions:**

Whether $d$ is identifiable in the nonlinear setting? And how to identify $d$ for the contrastive method?

I would like to raise my score if this problem can be addressed properly.



**Limitations:**

No potential negative societal impact.

---

> ### Author Rebuttal · Authors · 2023-08-09
>
> We thank the reviewer for the review and their suggestions. We will address their concerns in order.
>
> **Regarding the identifiability of $d$:**
>
> Yes, the dimension $d$ is identifiable from the observational distribution in our setting for the following reason. The image $f(\mathbb{R}^d)=M\subset \mathbb{R}^{d'}$ is a submanifold of dimension $d$, i.e., it locally looks like a $d$ dimensional hyperplane, so its dimension can be identified from the dimension of the tangent space.  Put differently, the datapoints in a small neighborhood of a point $x$ generate essentially a $d$ dimensional linear space.
>
> In the setting considered in [2] that the reviewer highlighted, the latent dimension can be directly estimated from the rank since the mixing $f$ was assumed to be linear (which is a specialized setting). In general, when $f$ is non-linear as in most representation learning tasks, this task of estimating the latent dimension for complex data is much harder but also highly important and has been subject to intense study.
>
> For instance, in [1] the authors define a maximum likelihood estimator for the dimension of the data manifold around a data-point $x$ (it exploits that the volume of the $d$-dimensional ball scales like $r^d$ with the radius $r$).
> We show in the table below our estimates when applying this estimator (for the $k=50$ nearest neighbors) to our MLP setting with observed dimension $d'=100$
> and $n=10000$ samples where we average the estimator over 10 different data-points $x$. We report the mean over 10 different runs and the  reported error is the standard deviation.
>
> | Ground truth $d$    | Estimated $\hat{d}$ |
> | -------- | ------- |
> | $5$  | $5.3\pm 0.3$    |
> | $10$ | $9.8 \pm 0.6$     |
> | $20$    | $16.2 \pm 0.6$    |
>
> So for the settings we consider we can estimate the dimension experimentally, but, as one would expect, the problem becomes more difficult for high dimensions.
> To use our contrastive algorithm with unknown dimension, one could first use a standard estimator to estimate the dimension $d$ of the data manifold (as in [1]) and then apply the contrastive algorithm.
> We thank the reviewer for bringing up this point, and we will clarify in the paper, that $d$ is also identifiable.
>
>
> **Regarding multi-node interventions:**
>
> This is an interesting direction that is beyond the scope of our present work but we expect that new block identifiability results can be derived for multi-node interventions. We will be happy to add a discussion of this along with further future directions.
>
>
> We hope our response clarifies the reviewer's points of concern, especially the ones that led them to reduce their score. We're happy to address any further concerns and also welcome additional feedback on improving the paper.
>
> [1] Elizaveta Levina and Peter Bickel. Maximum Likelihood Estimation of Intrinsic Dimension,  NeurIPS 2004.
>
> [2] C. Squires, A. Seigal, S. Bhate, and C. Uhler. Linear causal disentanglement via interventions, ICML 2023.

---

> > ### Comment · Reviewer_RqDB · 2023-08-16
> > **Reply to rebuttle**
> >
> > Thanks for the clarifications. I had some negative impressions on the paper when I first read it, since the identifiability of the latent dimension was the first thing I was looking for in the theoretical results. But it was not mentioned at all.
> >
> > Overall, this is a solid work on an important problem. I am satisfied with the provided evidence. Therefore, I would raise my score.

---

### Official Review · Reviewer_uV6M · 2023-07-02

**Soundness:** 4 excellent
**Presentation:** 4 excellent
**Contribution:** 4 excellent
**Rating:** 8
**Confidence:** 4

**Summary:**

This paper aims to identify latent variables via a nonlinear mixing function interventional data. The authors prove strong identifiability results for unknown single-node interventions, extending previous work that focused on linear maps, polynomial mixing functions or paired counterfactual data. The paper proposes an interesting contrastive algorithm to identify the latent variables and evaluates its performance on some synthetic tasks and a simple image dataset modified from Ahuja et al [2023]. While there have been other very recent works that examine the interventional setting, this is the first I'm aware of that only requires $d$ interventions to recover $d$ latents with minimal constrains on the mixing function.

**Strengths:**

* This is the strongest theoretical result that I am aware of in causal representation learning. The gaussian assumption is obviously restrictive, but other than that, the paper proves identifiability for a very practical class of problems: they only need $d$ interventions to identify $d$ latents, and make no further restrictive assumptions on the mixing function beyond standard infectivity / diffeomorphism assumptions.
* The paper is very clearly written both in the main text and the appendix.
* The contrastive algorithm that they propose to implement their approach is interesting. It has some optimization issues which the paper is upfront about, but I'm still curious to see how it would work on larger problems.

**Weaknesses:**

I don't have many complaints, but I do have some nitpicks.
* In the introduction (particularly the first paragraph), the paper makes it sound like the generative process for the data is via a neural network (transformers & diffusion models in line 16, and a similar comment is made in line 124 in support of Assumption 1). While we can build generative models of complex data with neural networks, that is not the $f(\cdot)$ that we care about in causal representation learning. The real generative function, $f(\cdot)$, is a property of the world---it's the camera or microscope that photographs a scene---and we don't have control over that. I realize this makes the diffeomorphism assumptions unrealistic, but I think it's better to just be upfront about that as a limitation.
* I have a similar complaint about the defence of the Gaussian assumption in lines 138 - 141: real processes almost certainly are not Gaussian, so we should treat it as a model of the world and be upfront about that (in the Box, "all models are wrong" sense). The paper would be strong if you were upfront about the fact that it is clearly restrictive but useful (because it gives strong identifiability results), and then evaluated sensitivity of the method to non-gaussian latents in the experiments.

More minor:
* Line 32 - I believe causal representation learning, but it remains to be seen whether it is either necessary or sufficient to build trustworthy systems, so that's a strong claim to make.

**Questions:**

* Have you experimented with non-gaussian latents? What happens?
* How well does the method work on larger problems (e.g. 5 - 10 balls? Or more?)

**Limitations:**

As mentioned above, the paper could be a little more upfront about the implication of the assumptions it makes, but overall it does a good job of addressing limitations. I also liked the "deadends" section at the end of the appendix.

---

> ### Author Rebuttal · Authors · 2023-08-09
>
> We thank the reviewer for their positive review and are glad the reviewer likes both our tight identifiability results and the contrastive learning algorithm. We agree with the reviewer's suggestions to improve the exposition and are happy to be revise the wording accordingly, including being more upfront about the limitations.
> In particular we will clarify that $f$ is a property of the world that we try to learn and Gaussian variables are often a useful approximation.
>
> We also like the reviewer's suggestions for additional experiments.
>
> 1. In Table 1 of the attached PDF, we show the results for experiments with non-Gaussian distributions (the setting agrees with the first row
> of Table 2 in the paper), in particular Laplace, Gumbel, Uniform and Exponential distributions.
> We observe that the recovery with our contrastive learning algorithm gets slightly worse but is still very reasonable. This is in line with the performance of, e.g., causal discovery algorithms based on a Gaussian likelihood score for non-Gaussian data.
>
> 2. As suggested, we also scaled up our experiments on the image dataset to 10 balls, please see Table 3. We found that for (much) larger sample size and larger models the performance can be increased significantly (compared to the results we reported), and we can handle  up to 10 balls. While there is certainly still room for improvement (e.g. via hyperparameter turning), the more challenging next step is to handle noisy observations and more complex sceneries.
>
> We will add these tables to the paper in the final version. We welcome additional feedback to improve the work.

---

> > ### Comment · Reviewer_uV6M · 2023-08-17
> > **...**
> >
> > Thank you for the additional experiments! Like reviewer 2jDD above, I will keep my score of 8, and am willing to advocate for the paper during reviewer-AC discussions.

---

### Official Review · Reviewer_2jDD · 2023-07-03

**Soundness:** 4 excellent
**Presentation:** 4 excellent
**Contribution:** 4 excellent
**Rating:** 8
**Confidence:** 5

**Summary:**

The paper considers the task of causal representation learning (causal disentanglement) from interventions, with (1) a linear latent structural causal model, (2) a nonlinear mixing function, and (3) single-node interventions. They show that, under perfect interventions, the generative model is identifiable up to trivial indeterminacies. They propose a method based on contrastive learning for recovering the generative model, and show that their method works well in practice (outperforming a baseline which only allow for linear mixing).

**Strengths:**

### Significance
This paper presents a significant theoretical contribution to an established line of work, while introducing techniques and connections that will be useful for future works. In particular, the advance from linear (or polynomial) mixing to general non-linear mixing is a big step towards realistic generative modeling. On the practical side, they also develop a solid contribution by showing that contrastive learning can be used to disentangle the latent variables.

### Clarity
The paper is very clear in describing its contribution and how it compares to previous work. The presentation is quite thorough: they clearly describe their setup and discuss their assumptions, they show that their sufficient conditions for identifiability are actually necessary conditions, they discuss the unsuitability of polynomial mixing, and they provide a detailed description of their experimental methodology with accompanying code.


**Weaknesses:**

There are no major weaknesses. There are some relatively minor points of confusion / potential typos and some small suggestions to improve the paper in the **Questions** section.

If I had to pick a weakness, it would be that the proposed algorithm suffers from problems with local optima. It would be ideal to have an algorithm which provably recovers the latent representation, and some sample complexity results. These results would make the paper feel very "complete", and with these results the paper would probably be a better fit for a journal.

**Questions:**

### Questions
1. In Equation (3), why are we able to identify shifts without scaling indeterminacy? This seems odd: if we scale a variable, I would expect that the shift also scales.
2. In Equation (7), should $h$ in the last term have a subscript? Then it is not a matrix so I don't see how it should be in the inner product.

### Suggestions
1. In line 26, the cited papers are about independent component analysis, and these are given as examples of identifiability in causal representation learning (CRL). I agree that ICA can be cast as a special case of CRL. However, I have the feeling that for the purposes of clarity, it might be best to use a different umbrella term like "identifiable representation learning"; there is nothing really "causal" about ICA.
2. Adapt the synthetic data generation process to reduce varsortability and $R^2$-sortability [1,2]. One procedure which takes these issues into account is given by [3].

[1] Reisach, A., Seiler, C., & Weichwald, S. (2021). Beware of the simulated DAG! Causal discovery benchmarks may be easy to game.

[2] Reisach, A. G., Tami, M., Seiler, C., Chambaz, A., & Weichwald, S. (2023). Simple sorting criteria help find the causal order in additive noise models.

[3] Squires, C., Yun, A., Nichani, E., Agrawal, R., & Uhler, C. (2022). Causal structure discovery between clusters of nodes induced by latent factors.

**Limitations:**

The authors have adequately addressed the limitations of their work.

---

> ### Author Rebuttal · Authors · 2023-08-09
>
> We thank the reviewer for their review and positive feedback.
> We are glad the reviewer also expects our ideas to contribute towards generative modeling in the real world, as that is one of the stronger motivations behind our work.
>
> **Regarding the questions:**
>
> 1. We can identify $\eta$ exactly because the scaling is absorbed in $B$. Or, in other words, $\eta$ is the amount of shift in the normalized Gaussian (this is evident in equation (2)) and therefore can be identified without scaling indeterminacy.
>
> 2. There should be no subscript, thanks for pointing out our typo.
>
> **Regarding the suggestions:**
>
> We appreciate the insightful suggestion regarding varsortability and will be happy to discuss this in the paper, as well as cite the works you linked.
> We ran an additional experiment where we standardized $Z$ before applying the non-linearity thereby removing the varsortability (we also checked that in our setting $R^2$-sortability does not deviate substantially from $1/2$). The results are in Table 2 in the attached PDF and show a slightly degraded performance.
> In general, note that it is not directly clear how varsortability of $Z$ can be exploited because $Z$ is only identifiable up to scaling, as we show.
>
> We agree with the reviewer that resolving the issues of local optima and sample complexity are of great interest. Although little progress has been made on them so far in this field
> (e.g.,  there are no sample complexity results for causal representation learning that we are aware of),
> these are important directions for further research.
> We plan to expand our discussion section to include those directions, and also incorporate your suggestions along with the suggestions by other reviewers.

---

> > ### Comment · Reviewer_2jDD · 2023-08-13
> >
> > I am pleased with the author's response to my review and have no remaining questions. I appreciate their consideration of the varsortability / $R^2$-sortability issue, and I agree that these problems are harder to conceptualize in the CRL setting.
> >
> > I will keep my score of 8, and am willing to advocate for the paper during reviewer-AC discussions.

---

### Official Review · Reviewer_LWnm · 2023-07-05

**Soundness:** 4 excellent
**Presentation:** 4 excellent
**Contribution:** 3 good
**Rating:** 7
**Confidence:** 4

**Summary:**

In recent years, the theory of non-linear independent component analysis and causal representation learning has witnessed a lot of interesting developments. In this work, the authors study the problem of causal representation learning in the presence of interventional datasets, where interventions occur on the latents. The authors show that when the latents follow a linear structural causal model with a Gaussian distribution, then under general non-linear mixing (diffeomorphisms) the latents can be recovered up to permutation and scaling. For imperfect interventions, the authors show that it is still possible to recover the partial order under topological ordering of the underlying causal graph G can be recovered.  The authors present a new method based on contrastive learning that learns to distinguish interventional data from observational data and test it out on some synthetic datasets.



**Strengths:**

The authors have studied an important problem in the area of intereventional causal representation learning. The results proposed in the work for the case of perfect interventions show that strong identification is achievable with perfect interventional data. The authors also make progress on the difficult problem of tackling imperfect interventions. The theory of the paper is overall quite insightful.

**Weaknesses:**

I have a question and a claim about weakness. Let us consider the following setting.

Data generation for obervational data is given as:

$\epsilon \sim \mathcal{N}(0, \sigma^2 Id)$,
$z = A \epsilon$,
$x  = f(z)$

where $A$ is invertible, $\mathcal{N}$ is normal distribution, $\epsilon$ is noise (each component is independent), $f(\cdot)$ is injective.

Data generation in interventional environment $k$ where $p^{th}$ component is intervened is given as

$\epsilon^{(k)} \sim \mathcal{N}(\mu_p e_p, \sigma^2 Id + \sigma_p^2 e_p e_p^{\top})$, $z^{(k)}= A \epsilon^{(k)}$, $x = f(z^{(k)})$

where $e_p$ is a vector with zeros everywhere except at $p^{th}$ entry, $Id$ is identity, $\mu_p$ is mean under intervened distribution

We index the observational data as $0$ and interventional data from $1$ to $p$.

If we condition on the index of the data, the different components of $\epsilon$ are conditionally independent and follow an exponential distribution.  We can now leverage the theory in i-VAE applied to identifying $\epsilon$ and mixing function $f \circ A$.  In this case, if we have $p\geq 2d$ and sufficient variability condition from Theorem 1 in http://proceedings.mlr.press/v108/khemakhem20a/khemakhem20a.pdf is satisfied, then we achieve permutation and scaling identification.


Therefore, we can assume to have identified $\epsilon$ up to permutation and scaling. We call this estimate $\hat{\epsilon} = P \Lambda z$, where $P$ is permutation, $\Lambda$ is diagonal. Note that $\hat{\epsilon} =\Lambda^{-1}P^{-1} A^{-1}z$. We now have a linear relationship between observed $\hat{\epsilon} $ and underlying true $z$.

We can now leverage imperfect intervention results under linear mixing from https://arxiv.org/pdf/2301.08230.pdf (Theorem 13, see Table 1) to achieve mixing consistency based identification of $z$.

We can even reduce the number of interventional distributions needed from $2d$ to $d$ provided we assume variance in Normal distribution is known by leveraging results from https://proceedings.mlr.press/v177/lachapelle22a/lachapelle22a.pdf.

The characterization I describe above suggests that it is possible to get identification results by combining i-VAE and  https://arxiv.org/pdf/2301.08230.pdf in a straightforward way. Either I am missing something? I would like the authors to clarify if they considered this simple combination? Further, if the authors agree with above characterization, how do the authors modify their contributions in this light?


I don't quite agree with line 253, distributional assumptions. I think Gaussians is still a strong assumption and authors would benefit by being upfront about it.

**Questions:**

Please see the weakness section above. My final score depends on clarifcation to the questions I raised above.

**Limitations:**

The authors would benefit by having a discussion on the limitations. For instance, they should talk about why is Gaussian assumption limiting as an example.

---

> ### Author Rebuttal · Authors · 2023-08-09
>
> We thank the reviewer for the review and their insightful suggestion of a simpler proof strategy.
> However, the model considered in their argument is the very special case when the causal graph's weights do not change at all under all considered interventions. In particular, their review assumes $z^{(k)} = A\epsilon^{(k)}$, but our setting is substantially more general and considers the case where $A$ changes (which covers more realistic interventions). We present additional details below.
> Regarding the distributional assumption, we agree it's needed in our work and it has been emphasized in the abstract and introduction. Additionally, we will carefully revise to make it more evident throughout the writeup.
>
>
> ==================
>
> **Additional details:**
>
> We follow your notation in this response, however, note that the matrix $A$ in this response corresponds to $B^{-1}$ in the paper.
> The review considers interventions of the form $z^{(k)}=A\varepsilon^{(k)}$ where $\varepsilon^{(k)}$
> is obtained by shifting and rescaling the noise variable of node $k$.
> As outlined in the review, this setting is covered by the iVAE theory and earlier identifiability results apply.
> Our work considers, in addition, interventions that change the relation to the parents, in particular perfect interventions that remove the effect from the parents.
> For such interventions, the matrix $A\to A^{(k)}$ also depends on the intervention.
> This is also the interventional setting considered for linear mixing functions in the recent work [1].
> Note that the settings considered by the reviewer and our work both agree in the special ICA case when there are no causal relations but don't agree in cases beyond ICA, which we handle in our work.
>
> Therefore, to achieve our tight identifiability results, we need to exploit the specific form of the interventions. Note that our proofs are not purely linear algebraic but also topological, i.e., we explicitly exploit continuity of the mixing (see Lemma 4) making it unlikely that our results directly follow from known theorems.
> Nevertheless, we agree that it is a valuable addition to our paper to clarify that the ICA case is substantially simpler and has been solved in earlier work.
>
> We hope that this changes your view on our theoretical contribution and other points that may have contributed towards a lower score, and we are happy to discuss this issue and all other questions further.
>
> [1] C. Squires, A. Seigal, S. Bhate, and C. Uhler. Linear causal disentanglement via interventions, ICML 2023.

---

> > ### Comment · Reviewer_LWnm · 2023-08-11
> > **Further thoughts**
> >
> > Thank you for the response. I have some follow up points.
> >
> > Firstly, I am not saying that your results from the above arguments I presented. However, I believe that the above case that I provided is both a very important one, complementary to your results, and follows from earlier work. I think you should reinterpret your contributions in the light of this example. For instance, you say in the abstract "This is also the first instance of causal identifiability from non-paired interventions for deep neural network embeddings." and have made several such statements elsewhere. The example I provided does not use paired interventions and also works for deep neural net embeddings.
> >
> > Each imperfect intervention can either occur by changing weights or noise distribution. The example I constructed shows that for latter class of imperfect interventions (that alter noise) you can use existing results to get stronger identification that you arrive. I don't think this result follows from your results either. Hence, the two results are complementary. Your results consider a larger class of imperfect interventions but arrives at weaker guarantees. The absence of any such discussion in the paper is not fair to earlier contributions that already exist.
> >
> > In your current response you seem to say "we agree that it is a valuable addition to our paper to clarify that the ICA case is substantially simpler and has been solved in earlier work." This is not exactly what I meant. I would appreciate if you can have a discussion of what I proposed  in the main body (with details in Appendix) of how i-VAE + recent linear mixing works such as Varici et al. solve important sub-cases.

---

> > > ### Author Response · Authors · 2023-08-14
> > > **Reponse to further thoughts**
> > >
> > > We thank the reviewer for their clarifying response.
> > > We apologize that we slightly misinterpreted  the criticism in your review.
> > >
> > > **tl;dr:** We are happy to include additional discussion on these points, which are quite subtle and likely require additional investigation to make work. Please see below for details.
> > >
> > > **Identifiability of the setting in the review:**
> > >
> > > Firstly, let's focus on the setting considered in the review and clarify why we say that it essentially only covers ICA and not general CRL.
> > > For the class of interventions that you consider, the causal structure encoded by $A$ cannot be identified
> > > and every causal graph is consistent with the observed distributions.  This can be seen as follows. As pointed out in the review, given the observational
> > > and interventional distributions $x^{(k)}$ we can find a function $g$ such that $x^{(k)}=g(\varepsilon^{(k)})$
> > > and $g$ is unique up to permutations and scale, so we can identify $\varepsilon^{(k)}$.
> > > However, we cannot identify $A$ (which is our goal).
> > > Indeed, given an arbitrary invertible matrix $\tilde{A}$ satisfying DAG constraints, we may define a new latent structure by $\tilde{z}^{(k)}=\tilde{A}\varepsilon^{(k)}$.
> > > Then $x^{(k)}=g(\varepsilon^{(k)}) =  g\circ \tilde{A}^{-1}(\tilde{z}^{(k)})$, and since $\tilde{A}$ was arbitrary, we get that the observations are compatible with any causal graph.
> > >
> > > This is consistent with Varici et al. [1] because in case of a linear SCM and interventions that only change the noise distribution (so the setting in the review), it seems that Assumption 3 in [1] is not satisfied for nodes with parents, i.e, the assumptions of their identifiability result are only satisfied for the empty graph. We will give more technical details at the end.
> > >
> > > Because of the counterexample above, it seems that to apply the techniques of [1] to the model considered in the review, we need to
> > >
> > > 1. either work in the special ICA setting
> > >
> > > 2. or make additional assumptions (as the reviewer seems to suggest).
> > >
> > > Finally, we note that our Theorem 3 applies to the setting in the review and implies identifiability of $\varepsilon^{(k)}$ up to linear maps. Therefore, we feel it's not fully complementary to this setting.
> > > Going further, we expect our linear identifiability result (Theorem 3) can be combined with any result
> > > for linear Gaussian SCMs and linear mixings such as [1]  to obtain, e.g., mixing consistency.
> > >
> > > ==========
> > >
> > > In conclusion, while the ICA case can be obtained from prior works, it is not clear to us how exactly identifiability results for nonlinear mixings beyond the case of independent latents can be obtained by combining known results.
> > > We kindly ask the reviewer to clarify whether we misunderstood or overlooked something or what additional data or assumptions are necessary.
> > > We think obtaining generalizations of Varici et al. [1] for non-linear mixing (using iVAE or otherwise), is a very interesting and nontrivial problem for future work, which [1] themselves pose as an open problem (in their conclusion section).
> > >
> > >
> > > As suggested by the reviewer, we will expand the prior work section with more details, and also revise the sentence claiming that this is the first CRL identifiability result for general nonlinear mixing because this is indeed a bit vague as there is no generally agreed upon definition of CRL.
> > > Thanks again for engaging in the discussion, and we are happy to clarify any further concerns.
> > >
> > > **Assumption 3 in [1] and pure noise interventions**
> > >
> > > Based on our understanding, for linear SCMs and interventions that only change the noise distribution (for the setting in the review),
> > >  Assumption 3 is only satisfied for interventions on nodes without parents, as outlined below.
> > >
> > > Denote the score (i.e., $\nabla \log (p)$)
> > > of the observational distribution $p_z$ by $s$ and the score for intervention $i$ with target $i$ by $s^i$.
> > > Then their Assumption 3 says that if $c\cdot (s - s^i)=0$ ($p_Z$ a.e.) for some vector $c$ then $c_j=0$ for $j\in \overline{pa}(i)=pa(i)\cup \{i\}$.
> > >
> > > We will assume that $i$  has a parent and show that the assumption is not satisfied.
> > > Assume that the structural equation for node $i$ is $Z_i = \alpha Z_{pa(i)} +N_i$ without intervention and under intervention $i$ the noise $N_i$ is replaced by $M_i$. Denote the log densities of $N_i$ and $M_i$
> > > by $n$ and $m$.
> > > Using Equations (6), (7) of [1], we get
> > >
> > > $s(z)-s^i(z)=\nabla (\ln(p(z_i|z_{pa(i)})) - \ln(p^{(i)}(z_i|z_{pa(i)})))
> > > =\nabla (n(z_i - \alpha z_{pa(i)}) - m(z_i - \alpha z_{pa(i)})).
> > > $
> > >
> > > Let $k\in pa(i)$ and denote its coefficient in the structural equation by $\alpha_k\neq 0$. Then the vector $c$ with $c_i=\alpha_k$, $c_k = 1$ and $c_j=0$ for $j\notin \{i,k\}$  satisfies
> > >
> > > $
> > > c \cdot(s(z)-s^i(z)) = \alpha_k (n'(z_i - \alpha z_{pa(i)}) - m'(z_i - \alpha z_{pa(i)}))
> > > -\alpha_k n'(z_i - \alpha z_{pa(i)}) +\alpha_k m'(z_i - \alpha z_{pa(i)})=0.
> > > $
> > >
> > > Thus Assumption 3 is not satisfied.
> > >
> > > [1] Varici et al., Score-based Causal Representation Learning with Interventions, 2023

---

> > > > ### Comment · Reviewer_LWnm · 2023-08-14
> > > > **Thanks a lot for the response**
> > > >
> > > > Thanks for the thoughtful comment. That is an extremely good point! So I was mistaken in thinking that i-VAE and Varici can be combined in a simple way to solve a subclass of the problems. If the above example that you point is right, then that is indeed not true.
> > > >
> > > > I am happy to raise my score to 7.

---

### Author Rebuttal · Authors · 2023-08-09

We thank all reviewers for their reviews pointing out that the paper 'presents a significant theoretical contribution' (R. 2jDD) that studies 'an important problem in interventional causal representation learning' (R. LWnm), proposes a 'novel and sound algorithm' (R. RqDB), and is 'very clearly written' (R. uV6m). Reviewers 2jDD and uV6m were quite positive about the paper, Reviewers RqDB and LWnm generally acknowledged the contributions of the paper but had questions regarding an alternative proof strategy and identifiability of the latent dimension.
Those are addressed in the individual responses.

Following the optional (and intriguing) suggestions of Reviewers uV6M and 2jDD, we ran additional experiments where we investigated
the effects of different noise distributions (to probe misspecification) and data standardization (to probe varsortability) on the algorithm's performance.
In addition, we considered more challenging settings for the balls dataset (scaling up from 3 balls to 10 balls).
The results can be found in the attached PDF, for additional details we refer to our responses to Reviewers
uV6M and 2jDD.

---

### Decision · Program_Chairs · 2023-09-21

**Decision:**

Accept (oral)

**Comment:**

This paper represents a significant contribution to the rapidly blossoming theory for identification of latent causal models. In particular, this result handles the setting of Gaussian latents and arbitrary mixing functions, while most prior works required specific parametric forms (e.g. linear or polynomial). As such, I (and, it seems, the reviewers) believe it has the potential for significant impact in the space.